# Unsupervised pose-aware part decomposition for 3D articulated objects

## Abstract

Articulated objects exist widely in the real world. However, previous 3D generative methods for unsupervised part decomposition are unsuitable for such objects, because they assume a spatially fixed part location, resulting in inconsistent part parsing. In this paper, we propose PPD (unsupervised Pose-aware Part Decomposition) to address a novel setting that explicitly targets man-made articulated objects with mechanical joints, considering the part poses. We show that category-common prior learning for both part shapes and poses facilitates the unsupervised learning of (1) part decomposition with non-primitive-based implicit representation, and (2) part pose as joint parameters under single-frame shape supervision. We evaluate our method on synthetic and real datasets, and we show that it outperforms previous works in consistent part parsing of the articulated objects based on comparable part pose estimation performance to the supervised baseline.

## 1 Introduction

Humans are capable of recognizing complex shapes by decomposing them into simpler semantic parts. Researchers have shown that infants learn to group objects into semantic parts using the location, shape, and *kinematics* as a cue (Spelke et al., 1995; Slater et al., 1985; Xu & Carey, 1996). Moreover, even very young infants can learn to reason about kinematics using non-sequential single frames (Shirai & Imura, 2014; Kourtzi & Kanwisher, 2000). Although humans can naturally achieve such reasoning, it is challenging for machines, particularly in the absence of a rich supervision.

Generative part decomposition and abstraction methods have a long-standing history in computer vision (Roberts, 1963; Binford, 1971). Learning to represent complex target shapes with simpler part components has a wide range of applications, such as structure modeling (Mo et al., 2020; Roberts et al., 2021) and unsupervised 3D part parsing (Chen et al., 2020; Paschalidou et al., 2021; Tulsiani et al., 2017). Previous studies have mainly focused on non-articulated objects. Because they exploit the consistent part location as a cue to group shapes into semantic parts, these approaches are unsuitable for decomposing articulated objects when considering the kinematics of *dynamic part locations*. In contrast, there exist discriminative approaches targeting man-made articulated objects for part segmentation, in addition to part pose estimation from single-frame input. However, they require explicit supervision, such as segmentation labels and joint parameters (Yi et al., 2018; Xiang et al., 2020; Li et al., 2020). Removing the need for such expensive supervision has been an important step toward more human-like representation learning (Becker & Hinton, 1992).

In this study, as a novel problem setting, we investigate the generative part decomposition task for man-made articulated objects with mechanical joints, considering part poses as part kinematics, in an *unsupervised fashion*. Specifically, we consider the revolute and prismatic parts with a 1 degree-of-freedom joint state as the part kinematics because they cover most of the kinematic types that common man-made articulated objects have (Xiang et al., 2020; Abbatematteo et al., 2020; Michel et al., 2015). This task aims to learn consistent part parsing as a generative shape abstraction approach similar to (Chen et al., 2019b) for man-made articulated objects with various part poses from single-frame shape observation. An overview is shown in Figure 1. This task expands the target of the current generative part decomposition's applications to articulated objects in novel ways, such as part pose consistent part segmentation and part pose estimation. To realize the task, we identify the two challenges; (1) for pose-aware part decomposition, the model must consider the kinematics

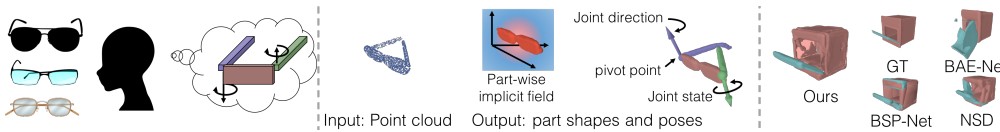

Figure 1: (Left) Even through independent observations, infants can build a mental model of the articulated object for part parsing based on its kinematics. (Middle) Likewise, we propose an unsupervised generative method that learns to parse the single-frame, unstructured 3D data of articulated objects and predict the part-wise implicit fields as well as their part poses as joint parameters. (Right) Our approach outperforms the previous works in consistent part parsing for articulated objects.

| | Part segmentation | Part pose estimation | Generative | Unsupervised |
|---|:---:|:---:|:---:|:---:|
| ANSCH (Li et al., 2020) | ✓ | ✓ | | |
| NASA (Deng et al., 2020b) | ✓ | | ✓ | |
| Nueral Parts (Paschalidou et al., 2021) | ✓ | | ✓ | ✓ |
| Ours | ✓ | ✓ | ✓ | ✓ |

Table 1: Overview of the previous works. We regard a method as unsupervised if the checked tasks can be learned only via shape supervision during training.

between possibly distant shapes to group them as a single part and (2) has to disentangle the part poses from shape supervision. A comparison with previous studies is presented in Table 1.

To address these challenges, we propose PPD (unsupervised Pose-aware Part Decomposition) that takes an unsegmented, single-frame point cloud with various underlying part poses as an input. PPD reconstructs part-wise shapes transformed using the estimated joint parameters as the part poses. We train PPD as an autoencoder using single-frame shape supervision. PPD employs category-common decoders to capture category-specific rest-posed part shapes and joint parameters. Learning to transform the rest-posed shapes properly disentangles shape and pose, and (2) restricting the position of the parts by the joint parameters forces shapes in distant space that share the same kinematics to be recovered as the same part. We also propose a series of losses, including an adversarial loss, to regularize the learning process. Furthermore, we employ non-primitive-based part shape representation and utilize deformation by part poses to induce part decomposition, in contrast to previous works that employ primitive shapes and rely on its limited expressive power as an inductive bias.

Our contributions are summarized as follows: (1) We propose a novel unsupervised generative part decomposition method for man-made articulated objects based on part kinematics. (2) We show that the proposed method learns a non-primitive-based implicit field as the decomposed part shapes and the joint parameters as the part poses, using single-frame shape supervision. (3) We also demonstrate that the proposed method outperforms previous generative part decomposition methods in terms of semantic capability (parsimonious shape representation, consitent part parsing and interpretability of recovered parts) and show comparable part pose estimation performance to the supervised baseline.

## 2 RELATED WORKS

Existing unsupervised generative part decomposition studies mostly assume non-articulated objects in which the part shapes are in a fixed 3D location (Tulsiani et al., 2017; Paschalidou et al., 2020; Chen et al., 2019b; 2020; Deng et al., 2020a; Kawana et al., 2020), or also targeting human body and hand shapes without considering part pose (Paschalidou et al., 2021). They induce part decomposition by limiting the expressive power of the shape decoders by employing learnable primitive shapes. Closest work of ours is BAE-Net (Chen et al., 2019b), whose main focus is consistent part parsing by generative shape abstraction. It also employs a non-primitive-based implicit field as the part shape representation, similar to ours. However, it still limits the expressive power of the shape decoder using MLP with only three layers. In contrast, our approach assumes parts to be dynamic with the consistent kinematics and induces part decomposition through rigid transformation of the reconstructed part shapes with the estimated part poses to make the decomposition pose-aware.

A growing number of studies have tackled the reconstruction of category-specific, natural articulated objects with a particular kinematic structure, such as the human body and animals. Representative works rely on the use of category-specific template models as the shape and pose prior (Loper et al., 2015; Zuffi et al., 2017; Bogo et al., 2016; Zuffi et al., 2019; Kulkarni et al., 2020). Another body

Figure 2: Model overview. To infer implicit field $\hat{O}$ based on part poses $\{B_i\}_{i=1}^N$ and part-wise implicit fields $\{\hat{O}_i\}_{i=1}^N$, the category-common decoders $F^{p,c}$ and $\{F_i^{s,c}\}_{i=1}^N$ capture pose biases and shape priors, the instance-dependent decoders $F^{p,z}$ and $\{F_i^{s,z}\}_{i=1}^N$ infer target specific components.

of works reconstruct target shapes without templates, such as by reconstructing a part-wise implicit field given a part pose as an input (Deng et al., 2020b) or focusing on non-rigid tracking of the seen samples (Božič et al., 2021). In contrast, our approach focuses on man-made articulated objects with various kinematic structures. Our approach learns the shape and pose prior during training, without any part pose information either as supervision or input, and is applicable to unseen samples.

In discriminative approaches, a number of studies have focused on the inference of the part segmentation of the input point cloud and part poses as joint parameters (Li et al., 2020; Xiang et al., 2020; Abbatematteo et al., 2020) targeting man-made articulated objects. These approaches require expensive annotations, such as part labels and ground-truth joint parameters. Moreover, they require category-specific prior knowledge of the kinematic structure. In contrast, our model is based on generative approach and is category agnostic. Moreover, it only requires shape supervision during training. A very recent work (Huang et al., 2021) assumes an unsupervised setting where multi-frame, complete shape point clouds are available for both input and supervision signals during training and inference. Whereas our approach assumes a single-frame input and shape supervision, it also works with partial shape input during inference. Note that, in this study, the purpose of part pose estimation is, as an auxiliary task, to facilitate consistent part parsing. It is not our focus to outperform the state-of-the-art supervised approaches in part pose estimation.

## 3 METHODS

In our approach, the goal is to represent an articulated object as a set of semantically consistent part shapes based on their underlying part kinematics. We represent the target object shape as an implicit field that can be evaluated at an arbitrary point $\mathbf{x} \in \mathbb{R}^3$ in 3D space as $O : \mathbb{R}^3 \to [0,1]$, where $\{\mathbf{x} \in \mathbb{R}^3 \,|\, O(\mathbf{x}) = 0\}$ defines the outside of the object, $\{\mathbf{x} \in \mathbb{R}^3 \,|\, O(\mathbf{x}) = 1\}$ the inside, and $\{\mathbf{x} \in \mathbb{R}^3 \,|\, O(\mathbf{x}) = 0.5\}$ the surface. Given a point cloud $I \in \mathbb{R}^{P_e \times 3}$ as an input, we approximate the object shape using a composite implicit field $\hat{O}$ that is decomposed into a collection of $N$ parts, where $P_e$ is a number of points in the point cloud. The $i$-th part has an implicit field $\hat{O}_i : \mathbb{R}^3 \times \mathbb{R}^{P_e \times 3} \to [0,1]$ as part shape and part pose $B_i \in SE(3)$. We ensure that $O$ is approximated as $O(\mathbf{x}) \approx \hat{O}(\mathbf{x} \,|\, I, \{B_i\}_{i=1}^N)$ through the losses.

An overview of PPD is shown in Figure 2. PPD employs an autoencoder architecture, and is trained under single category setting. Given a point cloud $I$, the encoder derives the disentangled shape latent vector $\mathbf{z}^s$ and the pose latent vectors $\mathbf{z}^p$ and $\mathbf{z}^{p,c}$. Category-common pose decoder $F^{p,c}$ captures joint parameter biases given $\mathbf{z}^{p,c}$. Instance-dependent pose decoder $F^{p,z}$ models residual joint parameters to the biases given $\mathbf{z}^p$. The part-wise category-common shape decoder $F_i^{s,c}$ captures category-common shape prior. Given $\mathbf{z}^s$ and conditioned by $F_i^{s,c}$, instance-dependent shape decoder $F_i^{s,z}$ infers residual shape details of the target shape to decode a part-wise implicit field $\hat{O}_i$. We discuss the details about $F^{p,z}$ and $F^{p,c}$ in Section 3.1, and $F_i^{s,z}$ and $F_i^{s,c}$ in Section 3.2.

### 3.1 PART POSE REPRESENTATION

We characterize part pose $B_i$ by its part kinematic type $y_i \in \{$fixed, prismatic, revolute$\}$ and joint parameters. Each $y_i$ is manually set as a hyperparameter. The joint parameters consist of the joint direction $\mathbf{u}_i \in \mathbb{R}^3$ with the unit norm and joint state $s_i \in \mathbb{R}^+$. Additionally, the "revolute" part has the pivot point $\mathbf{q}_i \in \mathbb{R}^3$. We refer to the joint direction and pivot point as the joint configuration. For the "fixed" part, we set $B_i$ as an identity matrix because no transformation is applied. For the

"prismatic" part, we define $B_i = T(s_i \mathbf{u}_i)$, where $T(\cdot)$ represents a homogeneous translation matrix given the translation in $\mathbb{R}^3$, and $s_i$ and $\mathbf{u}_i$ represent the translation amount and direction, respectively. For the "revolute" part, we set $B_i = T(\mathbf{q}_i)R(s_i, \mathbf{u}_i)$, where $R(\cdot)$ denotes a homogeneous rotation matrix given the rotation representation, and $s_i$ and $\mathbf{u}_i$ represent the axis-angle rotation around the axis $\mathbf{u}_i$ by angle $s_i$. In human shape reconstruction methods using template shape, its pose is initialized to be close to the real distribution to avoid the local minima (Kanazawa et al., 2018; Kulkarni et al., 2020). Inspired by these approaches, we parametrize the joint direction as $[\mathbf{u}_i; 1] = R(\mathbf{r}_i)[\mathbf{e}_i; 1]$, where $\mathbf{e}_i$ is a constant directional vector with the unit norm working as the initial joint direction as a hyperparameter and $\mathbf{r}_i \in \mathbb{R}^3$ represents the Euler-angle representation working as a residual from the initial joint direction $\mathbf{e}_i$. This allows us to manually initialize the joint direction in a realistic distribution through $\mathbf{e}_i$ by initializing $\mathbf{r}_i = \mathbf{0}$. Figure 3 illustrates the joint parameters.

Based on our observations, we assume that the joint configuration has a category-common bias, while the joint state strongly depends on each instance. This is because the location of each part and the entire shape of an object can constrain the possible trajectory of the parts, which is defined by the joint configuration. To illustrate this idea, we propose to decompose the joint configuration into a category-common bias term and an instance-dependent residual term denoted as $\mathbf{r}_i = \mathbf{r}_i^c + \mathbf{r}_i^z$ and $\mathbf{q}_i = \mathbf{q}_i^c + \mathbf{q}_i^z$,

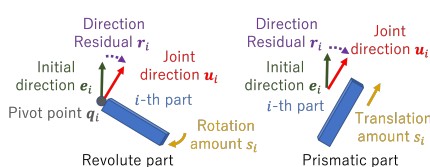

Figure 3: Geometric relationship between the joint parameters.

respectively. We employ the category-common pose decoder $F^{p,c}(\mathrm{qt}(\mathbf{z}^{p,c}))$, which outputs $\{\mathbf{r}_i^c \mid i \in \mathbb{A}^p\}$ and $\{\mathbf{q}_i^c \mid i \in \mathbb{A}^r\}$, where $\mathbb{A}^p = \{i \in [N] \mid y_i \neq \text{fixed}\}$, $\mathbb{A}^r = \{i \in [N] \mid y_i = \text{revolute}\}$, $\mathbf{z}^{p,c}$ denotes a pose latent vector, and $\mathrm{qt}(\cdot)$ is a latent vector quantization operator following VQ-VAE (Razavi et al., 2019). The operator $\mathrm{qt}(\cdot)$ outputs the nearest constant vector $\mathbf{c}^p$ to the input latent vector $\mathbf{z}^{p,c}$ among the $N_{qt}$ candidates. Instead of using a single constant vector, the model selects a constant vector among multiple constant vectors to capture the discrete, multi-modal category-common biases. We also employ an instance-dependent pose decoder $F^{p,z}(\mathbf{z}^p)$ that outputs $\{s_i \mid i \in \mathbb{A}^p\}$, $\{\mathbf{r}_i^z \mid i \in \mathbb{A}^p\}$, and $\{\mathbf{q}_i^z \mid i \in \mathbb{A}^r\}$. We constrain the possible distribution of the joint configuration around the category-common bias by the loss function explained in Section 3.3. This constraint incentivizes the model to reconstruct the instance-dependent shape variation by the joint state, which constrains the part location along the joint direction. This kinematic constraint biases the model to represent the shapes having the same kinematics with the same part. Because the previous works (Kawana et al., 2020; Deng et al., 2020a; Paschalidou et al., 2019) do not impose such a constraint on the part localization, learned part decomposition is not necessarily consistent under different poses.

## 3.2 PART SHAPE REPRESENTATION

We propose a non-primitive-based part shape representation that is decomposed into the category-common shape prior and instance-dependent shape details. We employ MLP-based decoders to model a part-wise implicit field. We capture the category-common shape prior using the category-common shape decoder $F_i^{s,c} : \mathbb{R}^3 \to \mathbb{R}$. Because $F_i^{s,c}$ does not take a latent vector from the encoder, it learns an input-independent, rest-posed part shape template as the category-common shape prior. We also employ an instance-dependent shape decoder $F_i^{s,z} : \mathbb{R}^3 \times \mathbb{R}^d \to \mathbb{R}$ to capture the additional instance-dependent shape details conditioned with the shape prior, where $d$ is the dimension of the shape latent vector $\mathbf{z}^s$. Given $F_i^{s,c}$ and $F_i^{s,z}$, we formulate a part-wise implicit field $\hat{O}_i$ as follows:

$$\hat{O}_i(\mathbf{x} \mid I) = \sigma(F_i^{s,z}(\mathbf{x}, \mathbf{z}^s)\hat{O}_i^c(\mathbf{x})) \tag{1}$$

where $\sigma(\cdot)$ represents the sigmoid function and $\hat{O}_i^c(\mathbf{x}) = \sigma(F_i^{s,c}(\mathbf{x}))$. For brevity, we omit $I$ in $\hat{O}_i$ and simply denote it as $\hat{O}_i(\mathbf{x})$. Given the part poses $\{B_i\}_{i=1}^N$ as part-wise locally rigid deformation, we formulate $\hat{O}$ as the composition of $\{\hat{O}_i\}_{i=1}^N$ defined as $\hat{O}(\mathbf{x} \mid I, B) = \max_i\{\hat{O}_i(B_i^{-1}\mathbf{x})\}$. As in the piecewise rigid model of (Deng et al., 2020b), coordinate transformation $B_i^{-1}\mathbf{x}$ realizes locally rigid deformation by $B_i$ of the part-wise implicit field by querying the rest-posed indicator. Note that, although we set the maximum number of parts $N$, the actual number of parts used for reconstruction can change; it is possible that some parts do not contribute to the reconstruction because of the $max$ operation or simply because $\hat{O}_i < 0.5$ for all 3D locations. In Equation 1, we experimentally found that conditioning $F_i^{s,z}$ by $\hat{O}_i^c$ through multiplication rather than addition effectively prevents $F_i^{s,z}$ from deviating largely from $F_i^{s,c}$. This regularization induces the unsuperivsed part

decomposition. Considering reconstructing the target shape by single $i$-th part, since the multiplication makes it difficult to output shapes that deviating largely from the category-common prior shape, the large shape variations of target shapes are expressed by $B_i$ regarded as the global pose of the reconstructed shape. However, the datasets' large shape variations in target shapes are due to the various local poses of multiple part shapes. Therefore, the large shape variations of target shapes cannot be expressed only by the single part and its part pose $B_i$. Thus, as an inductive bias of the unsupervised part decomposition, the model is incentivized to use a composition of multiple parts to express the shape variations due to various local part poses. In the learning process, the model first tries to reconstruct the target shapes with a single part; then with multiple parts. Lastly, it starts to deform each part to express the shape variations. During the learning process, the part poses are disentangled from the shape supervision to transform the part shapes in a way that minimizes the reconstruction loss. For the visualization of the learning process of part decomposition, see Figure 12 in the Appendix.

### 3.3 TRAINING LOSSES

**Shape losses.** To learn the shape decoders, we minimize the reconstruction loss using the standard binary cross-entropy loss (BCE) defined as:

$$L_{rec} = \lambda_{rec}\text{BCE}(\hat{O}, O) + \lambda_{rec}^c\text{BCE}(\hat{O}^c, O) \tag{2}$$

where $\hat{O}^c(\mathbf{x} \,|\, B) = \max_i\{\hat{O}_i^c(B_i^{-1}\mathbf{x})\}$, and $\lambda_{rec}$ and $\lambda_{rec}^c$ are the loss weights. The second term in Equation 2 is essential for stable training; it facilitates fast learning of $\{F_i^{s,c}\}_i^N$, so that $\{F_i^{s,z}\}_i^N$ can be correctly conditioned in the early stage of the training process. Moreover, because we consider the locally rigid deformation of the shape, the volumes of the shape before and after the deformation should not be changed by the intersection of parts; we formulate this constraint as follows:

$$L_{vol} = \lambda_{vol}(\mathbb{E}_{\mathbf{x}}[\text{relu}(\max_i\{F_i^{s,z}(\mathbf{B}_i^{-1}\mathbf{x}, \mathbf{z}^s)\})] - \mathbb{E}_{\mathbf{x}}[\text{relu}(\max_i\{F_i^{s,z}(\mathbf{x}, \mathbf{z}^s)\})])^2 \tag{3}$$

**Joint parameter losses.** For the joint parameters $\mathbf{q}_i$ and $\mathbf{r}_i$, we prevent an instance-dependent term from deviating too much from the bias term, we regularize them by the loss:

$$L_{dev} = \lambda_{dev}\left(\frac{1}{N^r}\sum_{i\in\mathbb{A}^r}\|\mathbf{q}_i^z\| + \frac{1}{N^p}\sum_{i\in\mathbb{A}^p}\|\mathbf{r}_i^z\|\right) \tag{4}$$

where $N^r = |\mathbb{A}^r|$, $N^p = |\mathbb{A}^p|$, and $\lambda_{dev}$ is the loss weight. Moreover, we propose a novel regularization loss that constrains the pivot point with the implicit fields. We assume that the line in 3D space, which consists of the pivot point and joint direction, passes through the reconstructed shape. The joint should connect at least two parts, which means that the joint direction anchored by the pivot point passes through at least two reconstructed parts. We realize this condition as follows:

$$L_{loc} = \frac{\lambda_{loc}}{N^r}\sum_{i\in\mathbb{A}^r}\left(\min_{\mathbf{x}\in\mathbb{S}_{gt}}\|\mathbf{q}_i - \mathbf{x}\| + \frac{1}{2}\left(\min_{\mathbf{x}\in\mathbb{S}_i}\|\mathbf{q}_i - \mathbf{x}\| + \min_{\mathbf{x}\in\mathbb{S}_{i,j}}\|\mathbf{q}_i - \mathbf{x}\|\right)\right) \tag{5}$$

where $\mathbb{S}_{gt} = \{\mathbf{x} \in \mathbb{R}^3 \,|\, O(\mathbf{x}) = 1\}$, $\mathbb{S}_i = \{\mathbf{x} \in \mathbb{R}^3 \,|\, \hat{O}_i(B_i^{-1}\mathbf{x}) > 0.5\}$, $\mathbb{S}_{i,j} = \{\mathbf{x} \in \mathbb{R}^3 \,|\, \hat{O}_j(B_j^{-1}\mathbf{x}) > 0.5, j \in \mathbb{A}^r \setminus i\}$, and $\lambda_{loc}$ is the loss weight. Note that $L_{loc}$ is self-regularizing and not supervised by the ground-truth part segmentation. See Figure 13 in the Appendix for an illustration of $L_{loc}$. To reflect the diverse part poses, we prevent the joint state $s_i$ from degenerating into a static state. In addition, to prevent the degeneration of multiple decomposed parts from representing the same revolute part, we encourage the pivot points to be spread. We realize these requirements by the loss defined as:

$$L_{var} = \frac{1}{N^p}\sum_{i\in\mathbb{A}^p}\left(\frac{\lambda_{var_s}}{\text{std}_{\mathfrak{B}}(s_i)} + \lambda_{var_q}\sum_{j\in\mathbb{A}^r\setminus i}\exp\left(-\frac{\|\mathbf{q}_i - \mathbf{q}_j\|}{v}\right)\right) \tag{6}$$

where $\text{std}_{\mathfrak{B}}(\cdot)$ denotes the batch statistics of the standard deviation, $v$ is a constant that controls the distance between pivot points, and $\lambda_{var_s}$ and $\lambda_{var_q}$ are the loss weights. Lastly, following the loss proposed in (Razavi et al., 2019), the pose latent vector $\mathbf{z}^{p,c}$ is optimized by the loss:

$$L_{vq} = \|\mathbf{z}^{p,c} - \text{sg}(\mathbf{c}^p)\| \tag{7}$$

where sg denotes an operator stopping gradient on the backpropagation.

**Adversarial losses.** Inspired by human shape reconstruction studies (Chen et al., 2019a; Pavllo et al., 2019), we employ the adversarial losses from WGAN-GP (Gulrajani et al., 2017) to regularize the shape and pose in the realistic distribution. The losses are defined as:

$$L_{adv_d} = \lambda_{adv_d}(\mathbb{E}_{\tilde{\boldsymbol{x}}\sim\mathbb{P}_g}[D(\tilde{\boldsymbol{x}})] - \mathbb{E}_{\boldsymbol{x}\sim\mathbb{P}_r}[D(\boldsymbol{x})]) + \mathbb{E}_{\hat{\boldsymbol{x}}\sim\mathbb{P}_{\hat{\boldsymbol{x}}}}\left[(\|\nabla_{\hat{\boldsymbol{x}}}D(\hat{\boldsymbol{x}})\| - 1)^2\right] \quad (8)$$

$$L_{adv_g} = \lambda_{adv_g}(-\mathbb{E}_{\tilde{\boldsymbol{x}}\sim\mathbb{P}_g}[D(\tilde{\boldsymbol{x}})]) \quad (9)$$

where $D(\cdot)$ is a discriminator; $\tilde{\boldsymbol{x}}$ is a sample from the reconstructed shapes $\mathbb{P}_g$ transformed by the estimated joint configuration and randomly sampled joint state $\tilde{s}_i \sim \text{Uniform}(0, h_i)$, with the maximum motion amount $h_i$ treated as a hyperparameter; $\boldsymbol{x}$ is a sample from the ground-truth shapes $\mathbb{P}_r$; $\hat{\boldsymbol{x}}$ is a sample from $\mathbb{P}_{\hat{\boldsymbol{x}}}$, which is a set of randomly and linearly interpolated samples between $\hat{\boldsymbol{x}}$ and $\boldsymbol{x}$; and $\lambda_{adv_g}$ and $\lambda_{adv_d}$ are the loss weights. As an input to $D$, we concatenate the implicit field and corresponding 3D points to create a 4D point cloud, following (Kleineberg et al., 2020).

### 3.4 IMPLEMENTATION DETAILS

We use the Adam solvers (Kingma & Ba, 2014) with a learning rate of $0.0001$ to optimize the losses: $L_{total_g} = L_{rec} + L_{vol} + L_{vq} + L_{dev} + L_{loc} + L_{var} + L_{adv_g}$ (sum of Equations 2, 3, 7, 4, 5, 6, and 9) and $L_{total_d} = L_{adv_d}$ (Equation 8), with a batch size of 18. For the input, we use the complete shape point cloud with 4096 points sampled from the surface of the target shape, unless otherwise noted. For the ground-truth implicit field, we use 4096 coordinate points and their corresponding indicator values. We set the loss weights as follows: $\lambda_{rec} = 0.01$, $\lambda_{rec}^c = 0.001$, $\lambda_{dev} = 0.1$, $\lambda_{var_s} = 0.1$, $\lambda_{loc} = 100$, $\lambda_{var_q} = 0.01$, $\lambda_{vol} = 1000$, $\lambda_{adv_g} = 0.65$, and $\lambda_{adv_d} = 0.35$. We set $v = 0.01$ in $L_{var}$ and $N_{qt} = 4$ for qt$(\cdot)$. For $h_i$ in $L_{adv_d}$, we set to $\frac{\pi}{2}$ and $0.4$ the "revolute" and "prismatic" parts, respectively. Note that we experimentally found that it does not constrain the model to predict $s_i$ larger than $h_i$ to reconstruct the target shape. Because we do not impose any geometric constraints on the part shapes, we set the number of parts for each part kinematics $y_i$ as its maximum number in the datasets plus an additional one part for over-parameterization. The detail of the datasets is explained in Section 4. We set $N = 8$, which consists of one "fixed" part, three "revolute" parts, and four "prismatic" parts. For the initial joint direction $\mathbf{e}_i$, for each "revolute" part, we set it to the $(+z, -z, +y)$ directions, and for each "prismatic" part, we set it to the $+x$ direction. We use the same hyperparameter for all categories, without assuming the category-specific knowledge. We train our network in two stages following (Chen et al., 2020): first, we train it on an implicit field of $16^3$ grids and then on $32^3$ grids. During the training, the max operation is substituted with LogSumExp for gradient propagation to each shape decoder. See Appendix B for further training details.

**Network architecture.** We use the PointNet (Qi et al., 2017)-based architecture from (Mescheder et al., 2019) as an encoder $E$ and the one from (Shu et al., 2019) as a discriminator $D$. Our shape decoders $\{F_i^{s,c}\}_{i=1}^N$ and $\{F_i^{s,z}\}_{i=1}^N$ are MLP with sine activation (Sitzmann et al., 2020) for a uniform activation magnitude suitable for propagating gradients to each shape decoder. For the category-common pose decoder $F^{p,c}$, we use two separate networks of MLP, namely, $F_r^{p,c}$ and $F_q^{p,c}$. For the instance-dependent pose decoder $F^{p,z}$, we employ MLP with a single backbone having multiple output branches. See Appendix A for further architectural details.

## 4 EXPERIMENTS

**Datasets.** Following the recent articulated pose estimation study (Li et al., 2020), we evaluate our method on five categories with various joint configurations from two synthetic datasets: Motion dataset (Wang et al., 2019) for the oven, eyeglasses, laptop, and washing machine categories, and SAPIEN dataset (Xiang et al., 2020) for the drawer category. Each category has a fixed number of parts with the same kinematic structure. We generate 100 instances with different poses per sample, generating 24k instances in total. We divide the samples into the training and test sets with a ratio of approximately 8:2. We also normalize the side length of samples to 1, following (Mescheder et al., 2019). For further details of the data generation, see Appendix C. To verify the transferability of our approach trained on synthetic data to real data, we use the laptop category from RBO dataset (Martín-Martín et al., 2018) and Articulated Object Dataset (Michel et al., 2015), which is the intersecting category with the synthetic dataset.

| | Drawer | Eye-glasses | Oven | Laptop | Washing machine | mean | # of parts |
|---|---|---|---|---|---|---|---|
| BAE (Chen et al., 2019b) | 6.25* | 11.11* | 73.06 | 25.11* | 80.30 | 39.17 | **1.42**/8 |
| BSP (Chen et al., 2020) | 66.31 | **70.69** | 81.65 | 76.68 | 87.92 | 76.65 | 27.50/256 |
| NSD (Kawana et al., 2020) | 38.39 | 42.11 | 74.67 | 74.44 | 89.11 | 63.75 | 10 |
| NP (Paschalidou et al., 2021) | 60.57 | 64.69 | **85.41** | 86.23 | 74.65 | 74.31 | 5 |
| Ours | **74.73** | 66.18 | 82.07 | **86.81** | **95.15** | **80.99** | 4.16/8 |

Table 2: Part segmentation performance in label IoU. Higher is better. The starred numbers indicate the failure of part decomposition and that only one recovered part represents the entire shape. The average and the predefined maximum numbers of recovered parts or primitives are shown before and after the slash, in the last column. *Ours uses more than one part on average with the least number.*

**Baselines.** We compare our method with the state-of-the-art unsupervised generative part decomposition methods with various characteristics: BAE-Net (Chen et al., 2020) (non-primitive-based part shape representation), BSP-Net (Chen et al., 2020) (primitive-based part shape representation with part localization by 3D space partitioning), NSD (Kawana et al., 2020) and Neural Parts (Paschalidou et al., 2021) denoted as NP (primitive-based part shape representation with part localization in $\mathbb{R}^3$). For BSP-Net, we train up to $32^3$ grids of the implicit field instead of $64^3$ grids in the original implementation to match those used by other methods. For NSD and Neural Parts, we replace its image encoder with the same PointNet-based encoder in our approach. For the part pose estimation, we use NPCS (Li et al., 2020) as the baseline. NPCS performs part-based registration by iterative rigid-body transformation, which is a common practice in articulated pose estimation of rigid objects. Note that NPCS assumes that part segmentation supervision are available during training and part kinematic type per part is known, which we do not assume in both cases. See Appendix B.1 for further training details of the baselines.

**Metrics.** For the quantitative evaluation of the consistent part parsing as a part segmentation task, we use the standard label IoU, following the previous studies (Chen et al., 2019b; 2020; Deng et al., 2020a; Kawana et al., 2020). As our method is unsupervised, we follow the standard initial part labeling procedure using a training set to assign each part a ground-truth label for evaluation purposes following (Deng et al., 2020a; Kawana et al., 2020). A detailed step can be found in Appendix D.1. For the part pose evaluation, we evaluate the 3D motion flow of the deformation from the canonical pose to the predicted pose measured as the endpoint error (EPE) (Yan & Xiang, 2016), which is a commonly used metric for pose estimation of articulated objects (Wang et al., 2019; Božič et al., 2021). We scale it by $10^2$ in experiment results. Finally, we report F-score, Chamfer L1 distance and volumetric IoU as the shape reconstruction accuracy metrics evaluated on the meshified implicit field sampled on $32^3$ grids using marching cubes (Lorensen & Cline, 1987). We use 100k points for the above three metrics, following (Mescheder et al., 2019; Deng et al., 2020a).

## 4.1 SEMANTIC CAPABILITY

We evaluate the semantic capability of our approach in part parsing. As part decomposition approaches aim to learn 3D structure reasoning with *as small a number of ground-truth labels as possible*, it is preferable to obtain the initial manual annotations with as *few numbers of shapes* as possible. This requirement is essential for articulated objects, which have diverse shape variations owing to the different articulations. As our approach is part pose consistent, we only need a minimal variety of instances for the initial manual labeling. To verify this, we evaluate the part segmentation performance using only the canonically posed (joint states were all zero) samples in the training set. See Appendix E.2 for further studies on pose variation for the initial annotation. The evaluation results are shown in Table 2. We also show the number of parts or primitives that each model uses in the last column of the table. Our approach outperforms all the previous works on average. The segmentation results are shown in Figure 4. The same color indicates the same segmentation part. The grey color of the part segmentation results of BSP-Net indicates that a primitive not labeled in the initial annotation is chosen to segment the surface area. Visualization procedure can be found in Appendix D.2. Our model uses a much smaller number of parts than BSP-Net (Chen et al., 2020); however, it still performs the best. This shows that our model is more parsimonious, and each part has more semantic meaning in part parsing. For additional visualization of our part segmentation result, see Appendix E.1. We also visualize the generated part shapes in Figure 5. We can see that

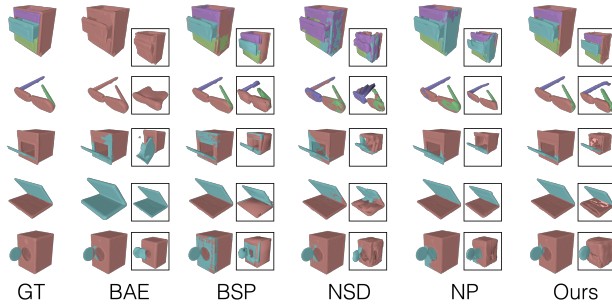

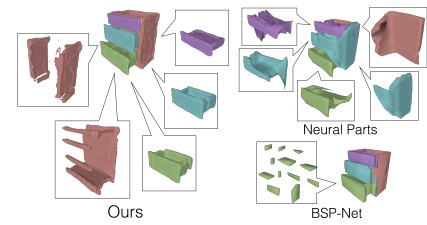

Figure 5: Visualization of parts and primitives. The boxes represent the parts or primitives used to reconstruct the semantic parts.

GT  BAE  BSP  NSD  NP  Ours

Figure 4: Visualization of the part segmentation. Mesh reconstruction is shown inside a box. Best viewed in zoom.

| | Drawer | Eye-glasses | Oven | Laptop | Washing machine | mean |
|---|---|---|---|---|---|---|
| NPCS (Li et al., 2020) (Supervised) | 1.598 | 1.087 | 2.702 | 0.751 | 1.594 | 1.546 |
| Ours (Unsupervised) | 3.452 | 2.631 | 3.360 | 2.546 | 2.529 | 2.903 |

Table 3: Part pose estimation performance in EPE. Lower is better.

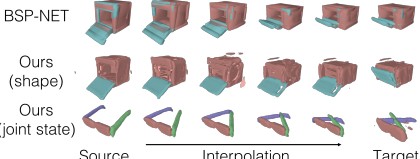

Figure 6: Interpolation in terms of shape and joint state.

one dynamic part is successfully reconstructed by a single implicit field. This demonstrates the advantage of using non-primitive-based part shape representation: complicated grouping mechanism of primitive shapes based on part kinematics is not needed. Also, our part shapes are more semantic and interpretable than the previous works. Moreover, our part shape representation exhibits the part shape with disconnected shapes, which the previous single primitive shape cannot express.

**Disentanglement between the part shapes and poses.** Because our approach disentangles shape supervision into part shapes and poses, it realizes pose-aware part decomposition. To verify the learned disentanglement, we visualize the interpolation results of part shapes and joint states as part poses in Figure 6. In the middle row, we show the shape interpolation between the source and the target while fixing the joint state $s_i$ of the source to maintain the same part pose. The shape is smoothly deformed from the source to the target maintaining the original pose. In the bottom row, we interpolate the joint state $s_i$ between the source and the target; the joint state changes from the source to the target maintaining the shape identity of the source shape. Our model successfully disentangles the part shapes and poses, unlike previous methods as shown in the top row.

## 4.2 PART POSE ESTIMATION

To validate whether the predicted part decomposition is based on the reasonable part pose estimation, we quantitatively evaluate the performance. Because we train our model without specifying a canonically posed shape, we use the estimated deformation between the target instance and the canonically posed instance of the same sample as the estimated part pose to align with the prediction of the supervised baseline. We present the evaluation results in Table 3. We show the supervised rigid registration approach NPCS (Li et al., 2020) only as a reference. Our method is comparable with NPCS, with the same order of performance. Note again that we are not attempting to outperform supervised pose estimation methods; rather, we aim to show that our unsupervised approach can decompose parts based on reasonable part pose estimation. See Appendix F for further results.

## 4.3 RECONSTRUCTION

We quantitatively validate whether PPD learns a reasonable shape representation rather than degenerated to ignore a target shape. We use BAE-Net (Chen et al., 2019b) as our baseline because it also focuses on part parsing through generative shape abstraction, and it also employs the same non-primitive part shape representation. The results are presented in Table 4. This study focuses on part parsing based on the challenging part pose estimation, thus persuing the state-of-the-art shape reconstruction accuracy as in (Chen et al., 2020; Kawana et al., 2020; Paschalidou et al., 2021) is out of focus. We show their results only as a reference. We can see that PPD outperforms BAE-Net.

|  | F-score ↑ | CD1 ↓ | IoU ↑ | # of params. | # of parts. |
|---|---|---|---|---|---|
| BSP (Chen et al., 2020) | 51.85 | 1.471 | 55.02 | 443.3 | 27.50/256 |
| NSD (Kawana et al., 2020) | 52.59 | 1.766 | 51.32 | 6.657 | 10 |
| NP (Paschalidou et al., 2021) | 46.35 | 1.85 | 43.09 | 24.43 | 5 |
| BAE (Chen et al., 2019b) | 24.83 | 3.211 | 33.67 | 52.50 | **1.42**/8 |
| Ours | **31.87** | **2.384** | **36.69** | **2.149** | 4.16/8 |

Table 4: Reconstruction performance. Chamfer L1 distance (CD1) and the number of parameters (# of params.) are scaled by $10^2$ and $10^5$, respectively. The average and the predefined maximum numbers of recovered parts or primitives are shown before and after the slash, in the last column.

|  | w/o $L_{vol}$ | w/o $L_{dev}$ | w/o $L_{loc}$ | w/o $L_{var}$ | w/o $L_{adv_g}$ | w/o CS | w/o CP | w/ all |
|---|---|---|---|---|---|---|---|---|
| Label IoU ↑ | 72.20 | 73.21 | 74.27 | 65.29 | 70.14 | 55.67 | 71.35 | **80.99** |
| EPE ↓ | 4.362 | 6.628 | 9.250 | 6.676 | 7.276 | 8.827 | 7.219 | **2.988** |

Table 5: Ablation study of the losses and the category-common decoders: "w/o CP" and "w/o CS" means disabling the category-common pose decoder and the shape decoders, respectively.

This is because PPD enables the use of a deeper network structure of shape decoders for better expressive power, and shape decoders are robust to unseen part poses owing to the disentangled part pose representation. For the further discussion on shape reconstruction, see Appendix G.

## 4.4 ABLATION STUDIES

We evaluate the effect of the proposed losses and the category-common decoders on part segmentation and part pose estimation. We disable each loss except $L_{rec}$ and $L_{vq}$ one at a time. We also disable the category-common decoders for pose and shape one by one and only use the corresponding instance-dependent decoder(s). The results are shown in Table 5. Enabling all losses and the category-common decoders performs the best. Particularly, disabling the category-common shape decoders significantly degrades both label IoU and EPE. This indicates that learning category-common shape prior is essential to perform proper part decomposition and to facilitate part pose learning, which is the core idea of this study. For further ablation studies, see Appendix H.

## 4.5 DEPTH MAP INPUT AND REAL DATA

Because PPD's decoders do not assume a complete shape as an input, it works with depth map input. Following BSP-Net (Chen et al., 2020), we train a new encoder that takes a depth map captured from various viewpoints as a partial point cloud and replace the original encoder. We minimize the mean squared error between the output latent vectors of the original and the new encoders so that their output are close for the same target shape. The results are shown in Table 6. The depth map input performs comparably to the complete point cloud input. We also verify that our model trained on synthetic depth maps reasonably generalizes to real data, as shown in Figure 7.

## 5 CONCLUSION

We propose a novel unsupervised generative part decomposition method, PPD, for articulated objects considering part kinematics. We show that the proposed method learns the disentangled representation of the part-wise implicit field as the decomposed part shapes and the joint parameters of each part as the part poses, using single-frame shape supervision. We also show that our approach outperforms previous generative part decomposition methods in terms of semantic capability and show comparable part kinematics estimation performance with the supervised baseline. Finally, we confirm that our model also works on the depth map input and generalizes to real data.

|  | F-score ↑ | Label IoU ↑ | EPE ↓ |
|---|---|---|---|
| Complete | 31.42 | 80.99 | 2.903 |
| Depth | 28.99 | 80.65 | 3.203 |

Table 6: Comparison between the point cloud input types: complete shape and depth map.

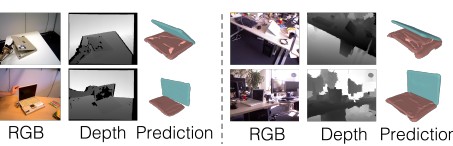

Figure 7: Real depth map input. (Left) RBO dataset (Martín-Martín et al., 2018) and (Right) Articulated Object Dataset (Michel et al., 2015).

## REPRODUCIBILITY STATEMENT

For the reproducibility, this paper includes the detailed description of our network architecture in Appendix A, implementation details on the hyperparameters in Section 3.4 and additional training details in Appendix B including model parameter initialization steps. Not only our models, but we also describe the training details of the baseline models in Appendix B.1. We also report the detailed steps of the data preparation process for the synthetic datasets in Appendix C. We describe the further detail on the data split in Appendix C.1 and data generation steps as well as publicly available source code that we use to generate the data in C.2. For evaluation, we report the steps for the initial labeling process used to evaluate unsupervised part segmentation results in Appendix D.1.

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

## A   NETWORK ARCHITECTURE

In this section, we explain the detailed architecture of the proposed network. The network architectures of the neural networks employed in the proposed method are depicted in Figure 8. The squircle diagram represents tensors, where the first and the second numbers inside the parentheses indicate the channel and the number of points, respectively. The squircle without the parentheses indicates the scalar value. For the split operation, $N[X, Y]$ denotes the split operation of the input tensor to $N$ number of sliced tensors with $X$ number of channels with $Y$ points. For the square diagrams with square brackets, the first and the second numbers in the square brackets indicate the input and output channels, respectively. The green square diagrams indicate multiple identical subnetwork architectures. $P_e$ denotes the number of points in the input point cloud to encoder $E$. For the other notations, see Section 3.

We use the simple PointNet architecture in the author-provided code of (Mescheder et al., 2019) as the encoder $E$. For the normalization layer in $F_r^{p,c}$ and $F_q^{p,c}$, we have experimentally found that using instance normalization (Ulyanov et al., 2016) for $F_r^{p,c}$ and layer normalization (Ba et al., 2016) for $F_q^{p,c}$ achieves the best performance. For the joint state $s_i$, we multiply $\pi$ to $\{s_i \,|\, y_i = \text{revolute}\}$. For the discriminator $D$, we use the architecture based on the PointNet (Qi et al., 2017) implementation in the author-provided code of (Shu et al., 2019). The weight of each linear layer in our discriminator is normalized using spectral normalization (Miyato et al., 2018) for stable training.

## B   TRAINING DETAILS

In this section, we explain the implementation and training details of the proposed models. We train our models per category with the same hyperparameter configuration described in Section 3.4 for all categories. For the input, we use the point cloud with 4096 points sampled from the surface of the target shape during the training. Unless otherwise noted, we use the complete shape point cloud. We use a batch size of 18. For the ground-truth implicit field, for each sample in a batch, we use 4096 3D coordinate points and their corresponding indicator values sampled from either $16^3$ or $32^3$ grids, depending on the training stage. This multi-stage training strategy on grids with different resolutions is inspired by (Chen et al., 2020). We train our network on $16^3$ grids in the first training stage. In addition, we set $\mathbf{r}_i = \mathbf{r}_i^c$ in the first stage. Then, we set $\mathbf{r}_i = \mathbf{r}_i^c + \mathbf{r}_i^s$ in the second stage. We determine the number of iterations for each stage according to the reconstruction loss and to the visualization of the reconstructed shapes on the validation data. It takes 2 to 3 days to train one model on a single NVIDIA V100 graphics card with 16 GB of GPU memory.

**Model parameter initialization.**   We use a sine function as a nonlinear activation function and the weight initialization strategy proposed in (Sitzmann et al., 2020) in our shape decoders, as follows:

$$w \sim \mathrm{U}\left(-\sqrt{\frac{6}{\text{IN}}}, \sqrt{\frac{6}{\text{IN}}}\right)\frac{1}{30} \tag{10}$$

where IN is an input channel to a linear layer, U is a uniform distribution, and $w$ is an element of the weight of a linear layer. For a linear layer that takes 3D coordinates as an input, we do not scale the weight $w$ by $\frac{1}{30}$.

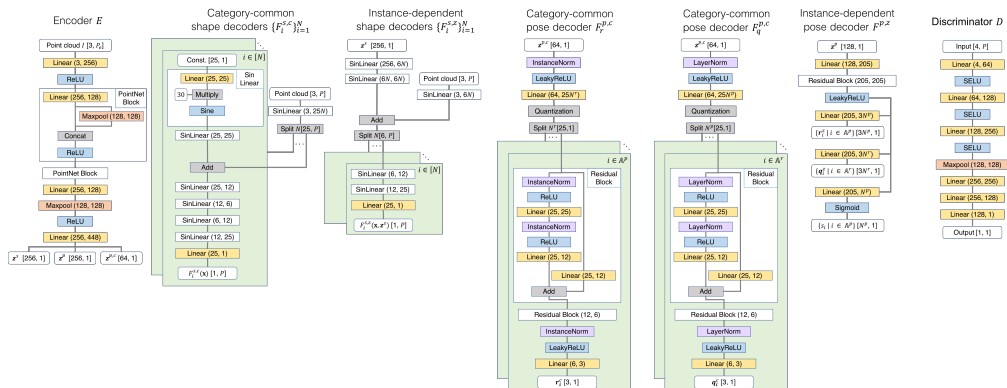

Figure 8: The architectures of our networks.

| | Drawer | Eye-glasses | Oven | Laptop | Washing machine |
|---|---|---|---|---|---|
| Training | 24 | 35 | 30 | 73 | 39 |
| Test | 6 | 7 | 7 | 13 | 6 |
| # of parts | (1 3 0) | (1 0 2) | (1 0 1) | (1 0 1) | (1 0 1) |

Table 7: Number of samples per category in each data split. Each sample is augmented by transforming its part pose to generate 100 instances. Numbers in a parethnesis in the last column indicates the ground truth number of fixed, prismatic and revolute type parts.

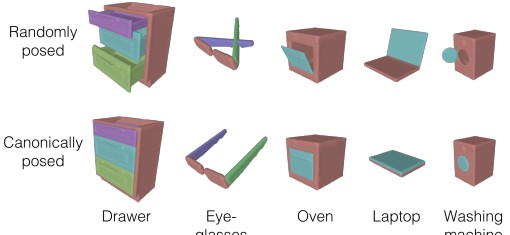

Figure 9: Visualization of the canonically posed and randomly posed ground-truth meshes of each category. The colors correspond to the different ground-truth part labels.

### B.1 TRAINING OF THE BASELINE MODELS

In this section, we describe the training details of the baseline methods. We use the author-provided implementations.

**BSP-Net (Chen et al., 2020).** Because the models in the author-provided codes of the other part decomposition baselines (BAE-Net (Chen et al., 2019b) and NSD (Kawana et al., 2020)) are trained on $32^3$ grids, we also trained BSP-Net on up to $32^3$ grids, compared to the $64^3$ grids in the original implementation. For training on the eyeglasses category, we could not successfully train the model even with different random seeds with the provided training script. After several trials, we experimentally found that scaling ground-truth indicator values by four for the first 20,000 iterations produced good initialization of the model. On the basis of this finding, we first pre-trained the model using the scaled ground-truth indicator values for 20,000 iterations for the eyeglasses category; then, we trained the model with the provided training script.

**NSD (Kawana et al., 2020) and Neural Parts (Paschalidou et al., 2021)** The model defined in the author-provided code takes an RGB image as an input, which is a more challenging setting for 3D shape reasoning than 3D shape input. We replace the image encoder of the original implementation with the same PointNet-based encoder used in our approach for a fair comparison.

**NPCS (Li et al., 2020).** In the experiment described in Section 4.2, we modified the original implementation of NPCS to use complete shape point clouds instead of partial point clouds of the depth map as an input with training from scratch, to remove the unnecessary performance degradation caused by pose ambiguity arising from the barely visible articulated part.

## C DATA PREPARATION

In this section, we describe our data preparation approach.

## C.1 DATA SPLIT

We split our training and test data according to the per-category data split approach introduced in (Li et al., 2020). We ensure that the test split contains at least six samples per category, except for the laptop category; therefore, the average split ratio is approximately 8:2. For the laptop category, we use 11 samples in the test split to make the split ratio comparable with those of the other categories. The number of samples in each split per category is presented in Table 7.

## C.2 GROUND-TRUTH IMPLICIT FIELD GENERATION

Following (Mescheder et al., 2019), we generate the ground-truth implicit field by the volumetric fusion of 100 depth images of a mesh object. For the mesh object, we sample 100 instances with randomly sampled part poses for each sample. For the pose sampling, we uniformly sample the rotation amount for each joint for the revolute joints. For the revolute joints of all categories except the eyeglasses category, we sample the rotation amount between $0°$ and $135°$. For the eyeglasses category, we sample between $0°$ and $90°$. For the prismatic joints of the drawer category, we sample the translation amount between 0 and the maximum amounts of the joints written in the URDF files of each sample in the SAPIEN dataset (Xiang et al., 2020). After we sample a part pose for each instance, we articulate the sample in its canonical pose (the rotation amount and translation amount were set to $0°$ and 0, respectively) using the sampled motion amount and ground-truth joint configuration. The canonically posed shape and the randomly posed shape of the same sample are shown in Figure 9. Finally, we normalize the size and location of the instances following (Mescheder et al., 2019). Specifically, we normalize the instances with the maximum extent collected from the instances generated from the same sample.

# D PART LABELING PROCEDURE FOR EVALUATION AND PART SEGMENTATION VISUALIZATION

## D.1 PART LABELING PROCEDURE

In this section, we explain the labeling procedure using the ground-truth part labels of the training samples to evaluate the part segmentation performance, following the same procedure used in (Kawana et al., 2020; Deng et al., 2020a). First, for each surface point sampled from the ground-truth part mesh of the instance of the training set, we determine the nearest reconstructed part and vote for the ground-truth part label of that point. Next, we assign each reconstructed part to the part label that has the highest number of votes. Finally, for each surface point sampled from the instance in the test split, we determine the nearest reconstructed part surface and assign the part label of the reconstructed part.

## D.2 PART SEGMENTATION VISUALIZATION

To visualize part segmentation, similar to (Tulsiani et al., 2017), We first measure the distance between a barycentric point of a ground truth mesh face to the surface of each part. Then we assign a mesh face the label of the part with the shortest distance to the barycentric point. Lastly, we color each face according to the obtained label.

# E SEMANTIC CAPABILITY

## E.1 ADDITIONAL VISUALIZATION OF THE PART SEGMENTATION

We visualize the additional part segmentation results of the proposed approach in Figure 10. Also, we visualize the part segmentation results given various part poses in Figure 11.

## E.2 PART SEGMENTATION USING ALL THE TRAINING SAMPLES

In Section 4.1, we show that our method works most efficiently by requiring instances with only a limited variety of poses for the initial annotations. In the experiment discussed in Section 4.1, we

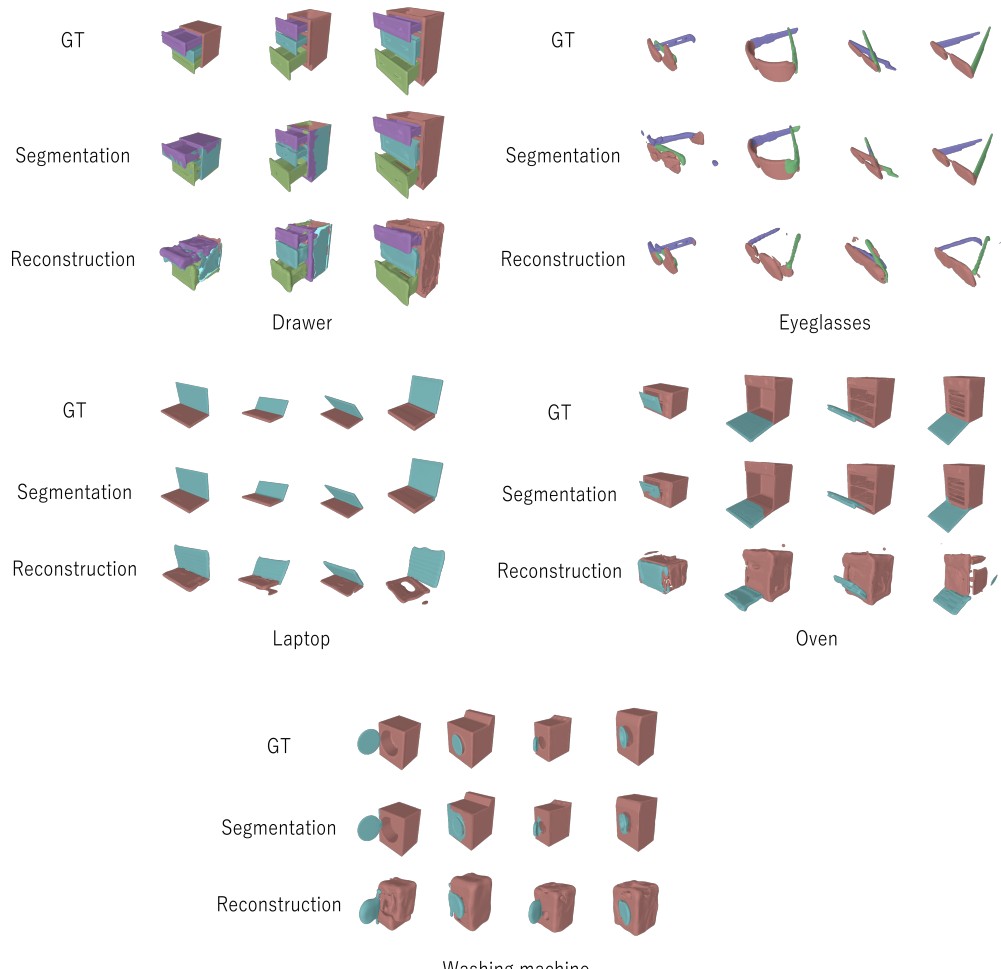

Figure 10: Visualization of the additional part segmentation results of the proposed approach with various samples. For drawer category, the different between some GT shapes are subtle (e.g., difference in handle shapes), we pick the three samples with distinct shape difference to avoid confusion.

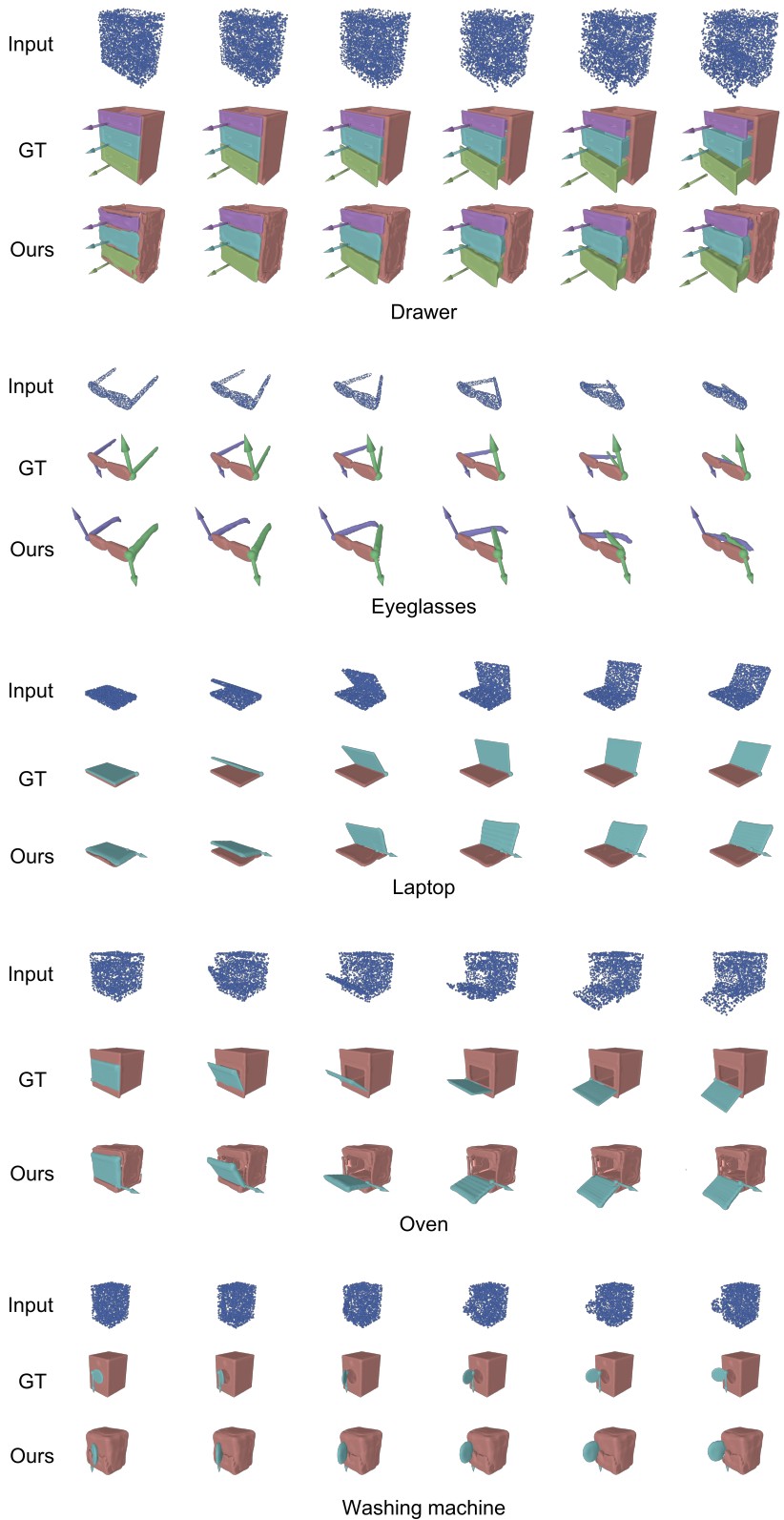

Figure 11: Visualization of the part segmentation results given input shapes with various part poses. Arrows in the figure indicate the ground-truth or predicted joint directions.

| | Drawer | Eye-glasses | Oven | Laptop | Washing machine | mean (All) | mean (Canonical) | Diff. (All - Canonical) | # of parts |
|---|---|---|---|---|---|---|---|---|---|
| BAE (Chen et al., 2019b) | 6.25 | 11.11 | 73.01 | 25.11 | 80.32 | 39.16 | 39.17 | -0.01 | **1.42**/8 |
| BSP(Chen et al., 2020) | 70.29 | **74.96** | **89.40** | 86.21 | **95.28** | **83.23** | 76.65 | 6.58 | 27.50/256 |
| NSD (Kawana et al., 2020) | 38.56 | 44.06 | 74.63 | 74.40 | 89.01 | 64.13 | 63.75 | 0.39 | 10 |
| NP (Paschalidou et al., 2021) | 60.56 | 64.75 | 85.33 | 86.22 | 74.72 | 74.32 | 74.31 | **0.01** | 5 |
| Ours | **74.83** | 66.25 | 82.06 | **86.80** | 95.18 | 81.02 | **80.99** | 0.04 | 4.16/8 |

Table 8: Part segmentation performance. We use all the instances in the training set to assign a label to each part as well as to the primitives. "Canonical" denotes the mean label IoU only using the canonically posed instances of the training for the label assignment. "Difference" shows the performance drop from the setting that uses all the instances in the training set to the setting that uses only the canonically posed instances. The average and the predefined maximum numbers of recovered parts or primitives are shown before and after the slash, in the last column. Our method achieves the same level of the label efficiency with Neural Parts with higher part segmentation performance.

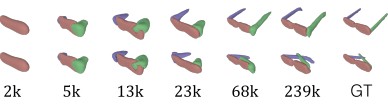

2k   5k   13k   23k   68k   239k   GT

Figure 12: Visualization of the training process. The first two rows show the reconstruction results for target shapes having the same part shapes but different part poses. The bottom row shows the number of training steps.

| | Drawer | Eye-glasses | Oven | Laptop | Washing machine | mean |
|---|---|---|---|---|---|---|
| BAE (Chen et al., 2019b) | 6.25* | 11.11* | 73.06 | 25.11* | 80.30 | 39.17 |
| BSP (Chen et al., 2020) | 26.62 | **71.14** | **85.19** | 64.41 | 86.60 | 66.79 |
| NSD (Kawana et al., 2020) | 34.07 | 60.06 | 70.09 | 70.12 | 62.97 | 59.46 |
| NP (Paschalidou et al., 2021) | 61.10 | 65.47 | 77.57 | 62.73 | 86.69 | 70.71 |
| Ours | **74.73** | 66.18 | 82.07 | **86.81** | **95.15** | **80.99** |

Table 9: Part segmentation performance in label IoU with the aligned number of parts for all methods ($N = 8$). The starred numbers indicate the failure of part decomposition and that only one recovered part represents the entire shape.

use canonically posed shapes, visualized in Figure 9, in the training set for the initial annotations. This section reports the evaluation setting where annotations of all training instances are available for the initial annotation, which is a favorable setting for the baselines. However, the annotation cost can be much higher in reality than in the previous setting. The results are shown in Table 8. Even under this setting favorable for the previous works, our method performs comparably with the state-of-the-art part decomposition method BSP-Net (Chen et al., 2020) using 256 primitives. It is not surprising that using many primitives achieves fewer part segmentation errors because, even when one primitive is inconsistently assigned to the ground-truth part, the impact on the label IoU is smaller. This is because a smaller portion of the evaluation points becomes erroneous compared with the model using fewer parts or primitives. Note that our research focuses on representing ground-truth articulated parts with consistently the same reconstructed parts by considering the part kinematics, unlike BSP-Net and the other baselines, which can assign different sets of primitives to the same articulated parts without considering the underlying part pose. To show the effectiveness of considering the part kinematics, we show the performance drop from using all training instances to using only the canonically posed instances in the table under the heading "Difference." We can see that our approach has the second best drop with the comparable number with Neural Parts (Paschalidou et al., 2021), yet higher part parsing performance. This shows that considering the part kinematics contributes to label efficiency by reducing the necessary initial annotation to perform well on the unsupervised part segmentation of articulated objects.

### E.3   PART SEGMENTATION WITH THE ALIGNED NUMBER OF PARTS

In this experiment we align the predefined maximum number of the parts of the baselines to same as ours ($N = 8$) to make the result more comparable. We change the predefined maximum number of primitives from 256 to 8 for BSP-Net (Chen et al., 2020), from 10 to 8 for NSD (Kawana et al., 2020), and 5 to 8 for Neural Parts (Paschalidou et al., 2021). Our method outperforms all the baselines on average with larger margin compared to the result in Table 2 of the main paper. The results are shown in Table 9. The smaller number of parts worsens NSD and BSP-Net's performance, and the opposite applies to Neural Parts.

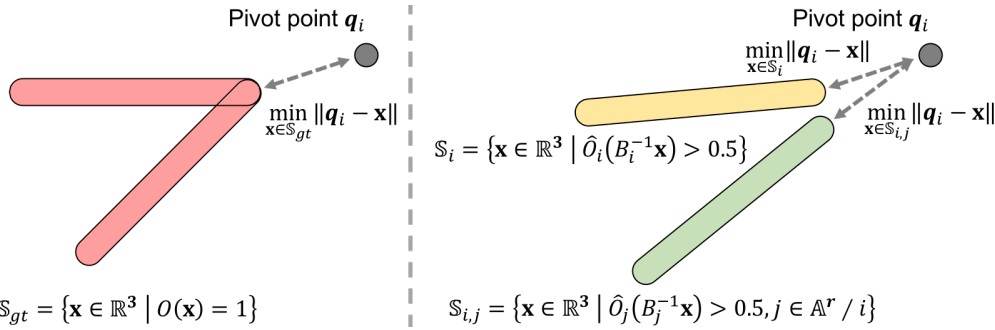

Figure 13: Illustration of $L_{loc}$ in 2D.

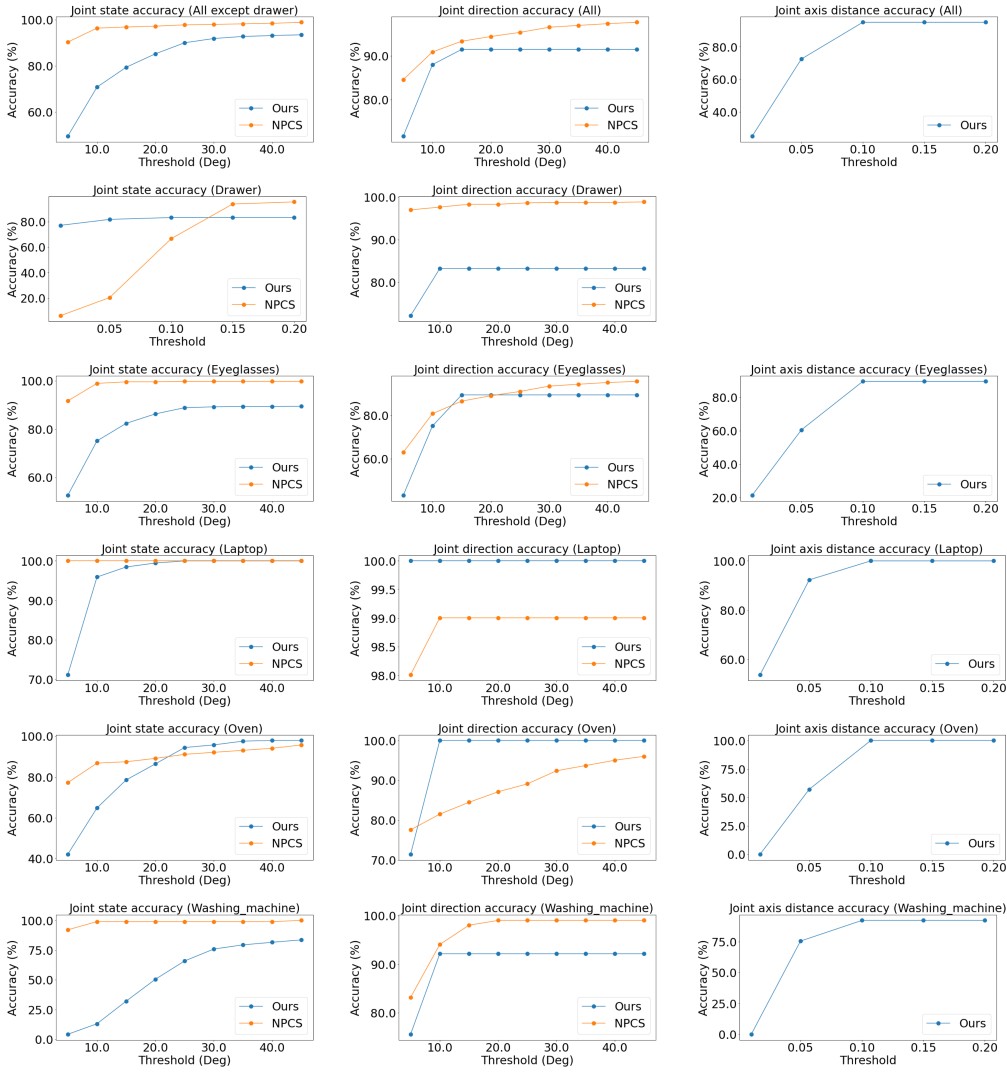

Figure 14: Joint parameter estimation performance.

| | Drawer | Eye-glasses | Oven | Laptop | Washing machine | mean |
|---|---|---|---|---|---|---|
| # of assigned parts | 1.0 | 1.0 | 1.0 | 1.0 | 1.0 | 1.0 |
| Part type accuracy | 89.50 | 83.25 | 100.0 | 92.14 | 100.0 | 91.46 |

Table 10: Evaluation results of the number of assigned reconstructed parts to the ground-truth parts and the part type accuracy. "# of assigned parts" shows the number of reconstructed parts assigned to the ground-truth parts, and "Part type accuracy" shows the percentage of part kinematic type matches between the ground-truth and the assigned reconstructed parts.

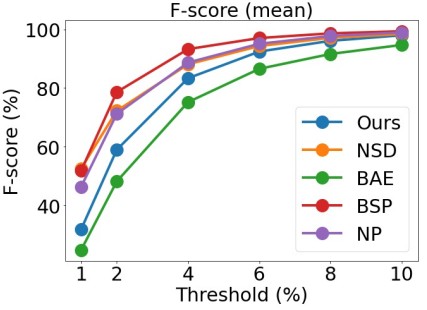

Figure 15: Plot of F-score on average with various thresholds.

## F    ADDITIONAL PART POSE EVALUATION

Because we train our model in an unsupervised fashion, through the labeling process described in Appendix D.1, the part kinematic types of the ground-truth and the assigned reconstructed part do not necessarily match. Moreover, multiple reconstructed parts may be assigned to one ground-truth part. Therefore, we choose EPE as the evaluation metric for part pose estimation due to its kinematic type agnostic property and calculation based on point correspondence between prediction and ground-truth, rather than part-level correspondence. In this section, as an additional part pose evaluation, we evaluate the accuracy of joint parameter estimation for "revolute" and "prismatic" parts. To avoid the problem of part pose evaluation in unsupervised learning described above, we evaluate the accuracy of joint parameter estimation by considering the prediction is correct when the following three conditions are all satisfied. (1) One reconstructed part is assigned to one ground-truth dynamic part. (2) The part kinematic type is the same between the ground-truth and the assigned reconstructed part. (3) The error of the joint parameters against the ground-truth is less than the error threshold. This evaluation method is more challenging than EPE because of the influence of (1) and (2) above, besides the prediction error of the joint parameters. We evaluate joint state accuracy and joint direction accuracy. Only for the revolute part,

we also evaluate joint axis distance accuracy, defined as the line to line distance between the ground-truth and the predicted line segments consisting of the pivot point and the joint direction. Figure 14 shows the evaluation results with varying error thresholds. We show the results of NPCS only as a reference; NPCS is a supervised model and assumes that the part segmentation is available during training, and the part kinematic types are also known. In contrast, our method learns both part segmentation and part kinematic type in an unsupervised fashion. Since NPCS does not estimate the pivot point, we only show the results of our method for joint axis distance accuracy. As for the joint state, we see reasonable accuracy of 70.80% for revolute parts on average when the threshold is less than 10 degrees and 79.43% when the threshold is 15 degrees. For the "prismatic" part of the drawer, our method outperforms the NPCS when the threshold is less than 0.1. For joint direction estimation, in three out of five categories (eyeglasses, laptop, and oven), our method is comparable or outperforming NPCS. In Table 10, we also show the number of reconstructed parts assigned to the ground-truth parts and the percentage of part kinematic type matches between the ground-truth and the assigned reconstructed parts. In all categories, the model correctly assigns one part. Moreover, even without part type supervision, our model successfully predicts correct part types with high accuracy of 91.46%. Improving the unsupervised learning of joint parameters under shape supervision is an interesting research direction.

## G    ADDITIONAL DISCUSSION ON SHAPE RECONSTRUCTION

Due to the space constraint in the main paper, we show category-wise shape reconstuction result in Table 11. Although it is not our focus to beat the state-of-the-art methods (Chen et al., 2020; Kawana et al., 2020; Paschalidou et al., 2021) in shape reconstruction accuracy, there are several possible reasons for the performance gap in shape reconstruction. The first reason is that our method relies on the challenging part pose estimation for shape reconstruction. Although our method estimates faithful part pose as evaluated in Section 4.2 and Appendix F, even the small pose error affects shape

| | | Drawer | Eye-glasses | Oven | Laptop | Washing machine | mean | # of params. | # of parts. |
|---|---|---|---|---|---|---|---|---|---|
| F-score ↑ | BSP (Chen et al., 2020) | 40.84 | 51.21 | 44.81 | 65.92 | 56.46 | 51.85 | 443.3 | 27.50/256 |
| | NSD (Kawana et al., 2020) | 48.63 | 50.7 | 37.56 | 83.58 | 42.5 | 52.59 | 6.657 | 10 |
| | NP (Paschalidou et al., 2021) | 43.50 | 52.68 | 31.52 | 72.21 | 31.84 | 46.35 | 24.43 | 5 |
| | BAE (Chen et al., 2019b) | 24.28 | 26.34 | 18.52 | 31.91 | 23.12 | 24.83 | 52.52 | **1.42**/8 |
| | Ours | **30.97** | **35.45** | **28.29** | **40.19** | **24.45** | **31.87** | 2.149 | 4.16/8 |
| CD1 ↓ | BSP (Chen et al., 2020) | 1.637 | 1.431 | 1.931 | 1.015 | 1.338 | 1.471 | 443.3 | 27.50/256 |
| | NSD (Kawana et al., 2020) | 1.594 | 1.642 | 2.819 | 0.649 | 2.128 | 1.766 | 6.657 | 10 |
| | NP (Paschalidou et al., 2021) | 1.958 | 1.528 | 2.533 | 0.823 | 2.399 | 1.848 | 24.43 | 5 |
| | BAE (Chen et al., 2019b) | 2.360 | 3.918 | 4.314 | 1.867 | 3.595 | 3.211 | 52.52 | **1.42**/8 |
| | Ours | **1.998** | **2.375** | **3.135** | **1.481** | **2.930** | **2.384** | 2.149 | 4.16/8 |
| IoU ↑ | BSP Chen et al. (2020) | 49.07 | 39.84 | 56.46 | 64.73 | 64.99 | 55.02 | 443.3 | 27.50/256 |
| | NSD Kawana et al. (2020) | 52.57 | 33.58 | 46.96 | 71.97 | 51.50 | 51.32 | 6.657 | 10 |
| | NP (Paschalidou et al., 2021) | 44.38 | 33.19 | 35.31 | 64.34 | 38.21 | 43.09 | 24.43 | 5 |
| | BAE Chen et al. (2019b) | 40.71 | 22.12 | 30.83 | 45.30 | 29.41 | 33.67 | 52.52 | **1.42**/8 |
| | Ours | **43.19** | **22.57** | **33.27** | **47.32** | **37.10** | **36.69** | 2.149 | 4.16/8 |

Table 11: Reconstruction performance. Chamfer L1 and the number of parameters (# of params.) are scaled by $10^2$ and $10^5$, respectively. The average and the predefined maximum numbers of recovered parts or primitives are shown before and after the slash, in the last column.

| | Label IoU ↑ | EPE ↓ |
|---|---|---|
| w/o $L^c_{rec}$ | 51.78 | 11.884 |
| w/o VQ | 72.78 | 10.772 |
| w/ all | **80.99** | **2.988** |

| | Drawer | Eye-glasses | Oven | Laptop | Washing machine | mean |
|---|---|---|---|---|---|---|
| # of used consant vectors | 3 | 4 | 4 | 3 | 3 | 3.4 |

Table 12: w/o $L^c_{rec}$ indicates disabling the second term of Equation 2 and w/o indicates disabling use of multiple constant vectors.

Table 13: The number of used constant vectors for the category-common pivot points $\{\mathbf{q}^c_i \,|\, i \in \mathbb{A}^r\}$. The maximum number is $N_{qt} = 4$.

reconstruction scores, especially for the dynamic part type. In Figure 10, the second sample of the eyeglasses category and the first sample of the oven category are good examples. Moreover, our method regularizes instance-dependent shape decoder's output as described in Section 3.2. This regularization results in less accurate shape reconstruction constrained by the shape prior than the state-of-the-art primitive-based shape representations. In Figure 10 for example, in the eyeglasses category, our model captures the global target shape variation yet struggles to recover the smaller details such as the front parts. To confirm how much reconstruction error those limitations could add to the performance gap with the state-of-the-art methods, we also evaluate F-score with various thresholds ranging from 0.01 to 0.1. Those thresholds can be intepreted as 1% to 10% distance error threshold for shapes whose side length is normalized to 1. The result is shown in Figure 15. On the reasonably moderate thresholds (4% to 6 %), our method reaches to the similar performance to the structured reconstruction methods. Improving reconstruction accuracy can be a important future work.

## H  ADDITIONAL ABLATION STUDY

As an additional ablation study, we also verify the effect of the second term of Equation 2 and the use of multiple constant vectors to model multi-modal category-common biases of joint configuration by vector quantization (Razavi et al., 2019) discussed in Section 3.1. The results is shown in Table 12. Disabling the second term of Equation 2 significantly drops the part segmentation performance and part pose estimation performance. Disabling the multiple constant vectors particularly affects the part pose estimation performance. Although we have found the category-common joint directions $\{\mathbf{r}^c_i \,|\, i \in \mathbb{A}^p\}$ tend to be encoded in the single constant vector, we have also found category-common pivot points $\{\mathbf{q}^c_i \,|\, i \in \mathbb{A}^r\}$ effectively utilizes the multiple constant vectors. We show the number of used consant vectors of the model reported in Table 2 in the main paper in 13. We can see that multiple constant latent vectors are used to decode the category-common pivot points.

