# OpenReview forum: "Unsupervised Pose-Aware Part Decomposition for 3D Articulated Objects"
_ICLR.cc/2022/Conference — ICLR 2022 Submitted_

### Official Review · Reviewer_YkRG · 2021-11-02

**Correctness:** 3
**Technical Novelty And Significance:** 4
**Empirical Novelty And Significance:** 3
**Recommendation:** 5
**Confidence:** 4

**Details Of Ethics Concerns:**

To the best knowledge of the reviewer, there is no ethics concerns about this paper.

**Main Review:**

Pros:

+ It is an unsupervised approach for parts decomposition which does not require parts labelling.
+ It can handle objects with varying number of parts.
+ It models the part poses of different kinematic types which can better model the articulated objects for unsupervised part decomposition.

Cons:
1)	It is not clear how to avoid the cases where multiple implicit fields are used to model the same part. Will the spread of the pivot points discourage the overlapping of the parts?
2)	It would be good to provide details about how to obtain the vector c^p.
3)	Please define the semantic meaning. It is not clear how each part is guaranteed to have semantic meaning.
4)	Is the parameterisation of the part pose correlated to the part semantics? In the implementation section, the number of different joint types are predefined. Will this implicitly indicate the parts structure? What is the influence of this predefined numbers on the segmentation results?
5)	What are the overlappings between different parts?
6)	It is not clear why not compare with Neural parts (Despoina et al, CVPR 2021) which is the most recent unsupervised parts segmentation method.
7)	The parts visualisation in Figure 4 is impressive. However, it is not clear how the drawers are segmented including the interior parts that are not visible from the surface defined by the point cloud.
8)	Please highlight the best performance in the Table 4. Based on the listed numbers, the proposed method does not achieve the best performance in terms F-score and Chamfer distance.
9)	It is not clear what the redundant parts learn if the parts number of the objects is smaller than the predefined maximum number.
10)	It would be interesting to visualise the category-specific shapes.
11)	Minors: The writing of the paper should be further polished. …poses can effectively addresses -> address;
Please explain semantic capability when it firstly appears in the text (above related works)
Please be consistent about the definition of {F_i^{s,c}}_{i=1}^N and {F_i^{s, z}} within the method section (see confusing definitions in the paragraphs above and below Sec. 3.1) Is {F_i^{s,c}}_{i=1}^N the category-common decoder or instance-dependent shape decoder?


**Summary Of The Paper:**

This paper addresses the problem of unsupervised parts decomposition for articulated objects with kinematic structures. In particular, the proposed approach disentangles the shape and pose of each part separately. Instead of adopting the primitive-based shape representation, each part is represented as an implicit field and disentangled part poses of three kinematic types. A category-specific shape/pose prior and instance-dependent shape/pose details are learned separately. The proposed approach is compared with existing works on benchmark datasets.

**Summary Of The Review:**

Overall, the proposed unsupervised parts segmentation approach from disentangled shape and pose is novel. However, the paper is not well-written. Details are not clear to the reviewer. The results are not quite promising as demonstrated in Table 4.

---

> ### Author Response · Authors · 2021-11-19
> **Response to reviewer#4 (1/2)**
>
> We thank reviewer#4 for pointing out some unclear points. Reference in the below comments can be found in the last post.
>
> > Based on the listed numbers, the proposed method does not achieve the best performance in terms F-score and Chamfer distance… The results are not quite promising as demonstrated in Table 4.
>
> Achieving the best reconstruction performance is not our focus in this research, rather our focus in Section 4.3 is to quantitatively confirm our method doesn’t just learn the meaningless shape. Also written in the responses to review#1 and section 2, and 4.3 of the main paper, our focus is on developing an unsupervised approach for consistent part parsing based on faithful part pose estimation to the novel object category that is man-made articulated objects. And we are happy to re-emphasize this in the updated draft. Especially reviewer#1 has supported, we believe that reconstruction performance alone does not diminish the main contribution of this paper. Unsupervised learning of consistent part parsing is one of the unique, important characteristics of part-based generative methods compared to the other shape reconstruction methods. Therefore, extending the part parsing capability to the novel category is more important than shape reconstruction accuracy alone. Also, our method successfully infers part poses besides part shapes, which adds another valuable part-wise information besides shape. We hope reviewer#3 will also value these contributions.
>
>
> > Is the parameterisation of the part pose correlated to the part semantics? In the implementation section, the number of different joint types are predefined. Will this implicitly indicate the parts structure?
>
> We do not think the predefined part types and joint parameter initialization per part indicate the part structure. Our predefined parts have redundant part types and numbers per part type. For example, the oven category does not have a prismatic type part, and the drawer category does not have revolute part types. Our predefined parts include revolute, prismatic, and fixed type parts with a redundant number, and which part type to be assigned to which GT part, and how many parts per part type are used is learned through training. We have shown that our model accurately infers part types and faithful joint parameters for GT dynamic parts in Table 9 of Appendix F.
>
> > … What is the influence of this predefined numbers on the segmentation results?
>
> We also would like to add an ablation study on different predefined primitive numbers in the camera-ready if accepted.
>
> > … Please define the semantic meaning ...
>
> Following the previous part decomposition studies, we define the semantic meaning as follows:
>
> 1. shape representation parsimony (using a small number of parts to describe target shapes) [1][2][3],
>
> 2. consistent part parsing ([1][2][3][4][5][6]),
>
> 3. interpretability of the decomposed part (e.g., shape interpretability [3][6]).
>
> Each item is verified as follows:
>
> 1. shape representation parsimony in the number of used primitives in Table 8 of Appendix E,
>
> 2. consistent part parsing in Section 4.1 and Appendix E,
>
> 3. interpretability of decomposed part shapes in Figure 4 and part poses in Section 4.2 and Table 9 in Appendix F.

---

> > ### Author Response · Authors · 2021-11-19
> > **Response to reviewer#4 (2/2)**
> >
> > > It is not clear how each part is guaranteed to have semantic meaning.
> >
> > Now we explain how we ensure semantic meaning explained above. To realize shape representation parsimony, our method uses a small number of parts compared to the baseline. To do so, we employ the non-primitive based shape representation which is able to faithfully reconstruct the GT part with complex topology with a few parts as illustrated in Figure 4. Using zero-genus primitive-based shape representation requires more number of parts to represent such non-zero-genus shape. For consistent part parsing, as we describe in the 4th paragraph of Section 3.2, we reconstruct the target shape by localizing the part-wise implicit field around the joint direction by predicted joint states. This incentivises the model to reconstruct the target part moving around the same joint direction (plus anchor point for revolute type part) with the same part, along with the mechanism to avoid the overlapping explained below. Lastly, we ensure interpretability by using the small number of parts along with consistent part parsing, so that each part reconstructs a meaningful shape and is easier to interpret. As shown in Table 9 in Appendix, the single part reconstructs the GT dynamic part. We can also interpret the pose of the part, as we explicitly infer it as the joint parameters.
> >
> > > It is not clear how to avoid the cases where multiple implicit fields are used to model the same part. Will the spread of the pivot points discourage the overlapping of the parts? … What are the overlappings between different parts?
> >
> > For the fixed type GT part, we think it’s acceptable for multiple implicit fields to model the same GT part. As we can see in Figure 4, the two reconstructed parts correspond to the same GT part. For the GT part with the dynamic part types (“revolute” and “prismatic”), we avoid overlapping by spreading pivot points. Moreover, the joint state is randomly initialized through the weight initialization of the pose decoder. This helps the different parts be transformed into different part poses by different joint states, thus avoiding the overlap.
> >
> > Moreover, the redundant, overlapping parts are degenerated through training and not reconstructed. We take max operation over the occupancy values of the multiple implicit fields as described in the second paragraph of Section 3.1 in the first draft. The max operation allows overlapping parts with lower occupancy values not to be reconstructed. Lastly, an implicit field of a part contributing reconstruction errors is learned to have lower occupancy values and gets degenerated. Thus it avoids such a part overlapping with a part that contributes to shape reconstruction.
> >
> >
> > > … it is not clear how the drawers are segmented including the interior parts ...
> >
> > Although input point clouds do not have interior shapes, GT shapes have in our experiment. Through the training, the model learns to infer the interior part shapes.
> >
> >
> > > … why not compare with Neural parts (Despoina et al, CVPR 2021) ...
> >
> > Also suggested by reviewer#2 and #3, we will add the comparison with Neural Parts [6] in the updated draft. We still outperforms Neural Parts in part segmentation (mean label IoU is 74.31%, and ours 80.99%).
> >
> >
> > Reference:
> >
> > [1] Shubham Tulsiani, Hao Su, Leonidas J Guibas, Alexei A Efros, and Jitendra Malik. Learning shape abstractions by assembling volumetric primitives. In Proceedings of the IEEE Conference on Computer Vision and Pattern Recognition (CVPR), pp. 2635–2643, 2017.
> >
> > [2] Despoina Paschalidou, Ali Osman Ulusoy, and Andreas Geiger.  Superquadrics revisited: Learning 3d shape parsing beyond cuboids. In Proceedings of the IEEE/CVF Conference on ComputerVision and Pattern Recognition (CVPR), pp. 10344–10353, 2019.
> >
> > [3] Yuki Kawana, Yusuke Mukuta, and Tatsuya Harada. Neural star domain as primitive representation. In Advances in Neural Information Processing Systems (NeurIPS), pp. 7875–7886, 2020.
> >
> > [4] Zhiqin  Chen,  Kangxue  Yin,  Matthew  Fisher,  Siddhartha  Chaudhuri,  and  Hao  Zhang.   Bae-net: branched autoencoder for shape co-segmentation. In Proceedings of the IEEE/CVF International Conference on Computer Vision (CVPR), pp. 8490–8499, 2019b.
> >
> > [5] Zhiqin Chen, Andrea Tagliasacchi, and Hao Zhang. Bsp-net: Generating compact meshes via binary space partitioning. In Proceedings of the IEEE/CVF Conference on Computer Vision and Pattern Recognition (CVPR), pp. 45–54, 2020.
> >
> > [6] Despoina Paschalidou,  Angelos Katharopoulos,  Andreas Geiger,  and Sanja Fidler.   Neural parts: Learning expressive 3d shape abstractions with invertible neural networks. In Proceedings of the IEEE/CVF Conference on Computer Vision and Pattern Recognition (CVPR), pp. 3204–3215,2021.

---

### Official Review · Reviewer_4Pds · 2021-11-02

**Correctness:** 3
**Technical Novelty And Significance:** 3
**Empirical Novelty And Significance:** 3
**Recommendation:** 8
**Confidence:** 2

**Main Review:**

STRENGTHS
- Well-written paper, detailed introduction, and motivation of the work.
- Good description of the method, with sufficient details to allow re-implementation and deep understanding of the implementation.
- Good evaluation, where the different tasks enabled by the method are evaluated: semantic segmentation, par pos estimation, and reconstruction. Also ablation studies are provided.
- First method to do unsupervised decomposition of articulated objects.

WEAKNESSES
- I think comparing to BAE and BSP can be a little bit misleading since these methods do not assume articulated objects. Working under the 'articulation' assumption allows the authors to frame their loss, and it is understandable that testing on articulated objects works better. A more fair (or additional) comparison would be to use this method on non-articulated objects and compare it to BAE and BSP.
- The symbols used for encoders and decoders are too cryptic and hard to parse while reading the paper (e.g. $F^{p,z}$, $F^{p,c}$, $F^{s,z}$, etc.). I'd encourage the authors to rename those to more meaningful names to ease reading.

**Summary Of The Paper:**

This submission proposes an unsupervised method to segment point clouds of objects that are articulated. In contrast to previous work with similar tasks, the learning is done in an unsupervised manner, outputting a segmented point cloud represented as a combination of implicit fields that can be rigidly articulated. The paper is well presented, technically sound, and correctly evaluated and compared to works with similar goals.

**Summary Of The Review:**

I think this is a nice written paper with a good technical contribution. The paper is well written and the results look convincing. Due to the specific scope of the paper (i.e., articulated objects) and the lack of works with this specific goal, comparisons to SOTA methods are somewhat difficult to parse. Nonetheless, seems to propose a step forward in the scenario that is designed for.

---

> ### Author Response · Authors · 2021-11-19
> **Response to reviewer#3**
>
> We first thank reviewer#3 for understanding the value of our study and supportive comments.
>
>
> > it is understandable that testing on articulated objects works better … A more fair (or additional) comparison would be to use this method on non-articulated objects and compare it to BAE and BSP.
>
> Our study particularly focuses on man-made articulated objects because they widely exist in the real world if no more than non-articulated objects, yet how to realize consistent part parsing based on part poses has not been addressed. And our approach is specifically designed for it. Therefore, there is not much benefit in applying our method to objects without articulation. Please note that our contribution is not just to have found that articulation assumption improves the part parsing but also to show how to learn it in an unsupervised fashion, including part poses, which is a previously unsolved setting.  We also interpret the intention of reviewer#3 as the current baselines do not claim to be applicable to articulated objects in their papers. Thus a more comparable baseline is required. Given this, in the updated draft, we will add Neural Parts [1] as a new baseline that claims to be applicable to articulated objects. The result shows that our approach still outperforms Neural Parts  (mean label IoU is 74.31%, and ours 80.99%).
>
>
> > The symbols used for encoders and decoders are too cryptic …
>
> We appreciate the suggestion to improve our paper, and we will simplify the symbols in the updated draft and the camera-ready if accepted.
>
> Reference:
>
> [1] Despoina Paschalidou,  Angelos Katharopoulos,  Andreas Geiger,  and Sanja Fidler.   Neural parts: Learning expressive 3d shape abstractions with invertible neural networks. In Proceedings of the IEEE/CVF Conference on Computer Vision and Pattern Recognition (CVPR), pp. 3204–3215,2021.

---

### Official Review · Reviewer_NczN · 2021-11-02

**Correctness:** 3
**Technical Novelty And Significance:** 3
**Empirical Novelty And Significance:** 3
**Recommendation:** 5
**Confidence:** 4

**Main Review:**

### Strengths:
------------
1. I really like the direction of having generative part-based models. I think this topic has not been adequately explored and while this paper does not provide enough experimental evidence on generative tasks, I consider it an important step towards developing generative models for part-based methods.
2. The paper seems technically sound and the proposed network architecture is novel.
3. I appreciate the authors's efforts in providing extensive experiments both in the main paper and in the appendix.

### Weaknesses:
-------------

1. My major concern for the paper is related to its clarity. While I overall like the proposed model, I found it quite hard to follow the paper. First of all, the goal of the proposed method is not crystal clear. In some parts of the text it is mentioned that the main focus of this work is developing a generative part-based model (e.g. 3rd paragraph in the Related Work Section), whereas elsewhere it is stated that the focus of this work is on learning shape abstractions (e.g. first sentence of Method Section). I do not think that the authors sufficiently demonstrate that the proposed method can be used for generative tasks, since they only show some interpolation results in Figure 5. However, this doesn't diminish the contributions of this work. Therefore, for future version of the paper, I would recommend that the authors try to ease their claims regarding their model being able to operate on generative tasks. Secondly, I think that the Method Section should be restructured in order to improve the paper's readability. Please see my comments below for more details. Finally, closely related to my first point, also the experiment section does not properly reflect the contributions of this work. Namely, just by looking Table 4, a reader would probably think that the proposed method is not the best choice if we are interested in accurate reconstructions, since it is consistently worse than all baselines wrt Chamfer loss for all object categories and also performs worse wrt F-score in the 4 out of 5 object categories.
2. My second concern is related to the dataset that the authors decided to use. I think the current dataset is quite simple. Since the authors are interested in articulated objects, it might be useful to also consider the D-FAUST dataset that contains humans performing various actions. I believe that the claims of the paper would be significantly stronger if the authors were showcasing that their model can also operate in more challenging setups.
3. While I appreciate the author's efforts in providing extensive experiments both in the main paper and in the supplementary. For the final version, I would like to see additional qualitative results on more object instances for the various categories. From the current experiments, it looks to me that the same objects instances are repeatedly used. Showing additional examples, would only make the claims of the paper stronger.

### Questions / Detailed Comments:
-------------------------------

1. For the experiment in Table 2, what is the number of ground-truth parts for each object category (on average)? Are these numbers reported in Table 7 in the Appendix? I think it would be better to fix the number of parts for all baselines (similar to Fig 3.) so that the results are comparable because with the current comparison it is unclear whether the superiority of the proposed model stems from the fact that it is using fewer components, which result in more coarse abstractions.
2. For the experiment in Table 4, can the authors clarify how is the F-score computed? In addition, how many points are used for computing the Chamfer-L1 loss? Is there any reason why the authors are not reporting the mean-IoU which is also commonly used for evaluating the reconstruction accuracy of primitive-based representations? I believe that including an additional comparison wrt the IoU would make this experiment more complete. Finally, it might also be useful to compare the proposed method with the more recent work of [2] since the authors demonstrate that they can capture complex geometries with few components.
3. For the experiment in Table 4, can the authors please clarify how the authors compute the number of learnable parameters for each method? Instead, I think it would be better, to report the number of primitives used by each method, as in Table 2.
4. Can the authors describe their intuition for the second term of Eq. 2. Is the second term really necessary? Unless I am misunderstanding something, I would expect that only using the first term of Eq. 2, which is the standard reconstruction loss for the majority of part-based models that rely on implicit surfaces, should yield good results? Can the authors elaborate on this and maybe also add a small quantitative comparison with and without the second term of Eq 2?
5. I am wondering whether the use of the VQ-VAE is really necessary. While I understand the authors's argumentation regarding its use allowing to capture discrete, multi-modal category common biases, I believe it would be beneficial for the paper to also ablate its use. Therefore, I recommend that the authors include this experiment for the final version of their paper.
6. In page 4, in Section 3.1, the authors state that their model first tries to reconstruct the target shape with a single part and as the training progresses it focuses on the remaining parts. This is also nicely shown in Figure 11 in the Appendix. Can the authors provide some intuition regarding why this is the case? For the majority of part-based methods, the models tend to distribute all parts on the target shape and then as the training progresses, each part focuses on a specific region of the target object. Why is this different for the proposed method? Moreover, if the model manages to capture the geometry of the target object using a single part, what is the incentive for utilizing multiple components in later training stages? I personally, find it quite hard to explain why this is the case.
7. Since the proposed method performs worse than all baselines wrt Chamfer loss and F-score (see Table 4), I believe the authors should try to describe why this is the case. Intuitively, I would have expected the proposed method to perform comparable to BSP-Net, however BSP-Net seems to be significantly better. My intuition is that for BSP-Net the authors are using significantly more primitives, thus the discrepancy. However, it would be nice to also elaborate more on this.

### Suggestions:
--------------

1. In page 3, in Section 3, in the second paragraph, where the authors briefly describe the components of their architecture, I think it would be helpful to add a sentence briefly summarizing the purpose of each component intuitively and at a higher level. Currently, from reading this paragraph it is hard to say what is the purpose of the having a category-common shape decoder and an instance-dependent shape decoder. Simply adding a sentence elaborating on the purpose of each component would significantly improve the paper's clarity. Similarly, I would really appreciate it if the caption of Figure 2, was more detailed. Specifically, it would be nice to provide the higher level concept of each component in order to allow the reader to better comprehend how the model works.
2. I think it would be great to move Figure 12 from the appendix to Section 3.2 in the main paper. The notation in this chapter is quite heavy and adding a figure that shows each variable would definitely be useful for the reader.
3. In the caption of Table 2, it would be better to mention what is metric that is reported in the corresponding table. In addition, in the table caption, instead of mentioning generated parts, I think it is more accurate to mention recovered parts.
4. In Section 4.1, the authors state "We can also see that our part shapes are more semantic and interpretable than the previous works.". I am not sure whether this is 100% true, since the final decomposition in particular for the case of NSD seems to be quite comparable with the proposed method.
5. I think that the Method Section can be restructured. I recommend to first describe the part pose parametrization (Section 3.2), the introduce the part shape representation through the implicit functions (Section 3.1) and finally add a separate section that presents and describes the loss terms of Eq. 2,3,4,5,6,7,8,9.
6. Throughout the text, the authors say that they have "single-frame shape supervision". Maybe instead say that the supervision comes in the form of a watertight mesh parametrized as a set of occupancy pairs indicating whether a point lies inside or outside the target object.

[1] Dynamic {FAUST}: Registering Human Bodies in Motion, Bogo et al. CVPR 2017

[2] Neural parts: Learning expressive 3d shape abstractions with invertible neural networks., Paschalidou et. al. CVPR 2021

### Minor Comments / Typos:
-------------------------
- In page 2, "To address the problems associated with (1) kinematically learning ... can effectively addresses these problems.": the authors should state how the above concerns are addressed with the current model.
- In page 3, "In contrast, our approach focuses on the man-made articulated object of the general category with various kinematic structures.": instead I would rephrase to "In contrast, our approach focuses on man-made articulated objects of a category with various kinematic structures."
- In page 3, "PPD employs an autoencoder architecture, and trained ...": instead it should be "is trained"
- In page 3, "Given a single-frame point cloud I ...": maybe remove "single-frame", I personally find it a bit confusing
- In page 5, "Through the observation, we assume ...": It is not clear to which observation the authors refer to.

**Summary Of The Paper:**

The proposed paper introduces a pose-aware part-based method for articulated objects. The proposed model takes as input a pointcloud and predicts a set of part poses as implicit functions. The network architecture comprises an encoder that maps the input pointcloud to three latent vectors, a category-common decoder and an instance pose decoder that take the two latent vectors and regress the part pose parameters and a set of shape decoders that are used for generating the part-wise implicit fields. To the best of my knowledge, this architecture is novel and the authors demonstrate that their model yields better segmentation results than various baselines on the Motion dataset. Moreover, the authors evaluate their model wrt the reconstruction accuracy of the final prediction. From the provided analysis it seems to perform worse than existing baselines. Finally, while the authors claim that their model can be used on various generative tasks, they only provide some interpolation results. To summarize, I think this paper is an important step towards pose-aware primitive-based representations, however some parts of the text require further clarifications.

**Summary Of The Review:**

While I like the paper, some aspects are unclear to me and I expect the authors to clarify them during rebuttal. Regarding the paper's clarity, I am confident that the paper can be significantly improved if the authors spend some time restructuring and rewriting parts of the text. Finally, I believe that showing additional qualitative results and ideally also providing results on a more challenging dataset would make the claims of the paper significantly stronger. Therefore, I think that the paper in its current state is below the threshold of acceptance. However, if the authors address my concerns I am happy to increase my initial score.

---

> ### Author Response · Authors · 2021-11-19
> **Response to reviewer#2 (1/3)**
>
> First of all, thank you for the very detailed suggestion to improve our paper; we will reflect on them in our updated draft. We will also organize unclear parts for a better understanding of the paper, as suggested.
> Reference in the below comments can be found in the last post.
>
> > … I would recommend that the authors try to ease their claims … on generative tasks.
>
> We first appreciate reviewer#2 understands the value and novelty of our study beyond a simple comparison of reconstruction accuracy.
>
> We are happy to update the first draft to more clearly state that our approach is for consistent part parsing by shape abstraction based on faithful part shape reconstruction and part pose estimation, and generative applications such as accurate reconstruction and novel shape generation are out of scope. As stated in Section 2 and Section 4.3 of the first draft, our focus is not beating the SoTA approaches in the reconstruction methods in the shape reconstruction performance. But instead, our focus is on consistent part parsing and on showing that considering part pose improves the part parsing performance, which we proved in Section Table2.
>
> > ...  the Method Section should be restructured …
>
> We will reorganize the section for readability, and we will update the draft accordingly. We promise we restructure the section in camera-ready at the latest if accepted.
>
> > … also the experiment section does not properly reflect the contributions of this work ...
>
> We will modify the reconstruction section to compare our work mainly with BAE-Net  because of the change mentioned in the first response that we will more explicitly state that this study is for part parsing by shape abstraction and part pose estimation. However, I also think the reader would wonder about the reconstruction performance compared to the recent part decomposition methods (BSP-Net, NSD, Neural Parts), we would like to list the result in the main paper if the space allows. Otherwise we will move the comparison with the recent part decomposition methods in the appendix. We think this change is valid because, in section 4.3, as we tried to indicate in the first paragraph, we intend to confirm our model learns reasonable shape representation on part parsing rather than claiming our approach for accurate shape reconstruction applications. Although our approach does not achieve SoTA reconstruction performance, as reviewer#2 agrees, we also believe this work is an important step towards developing generative models for part-based methods targeting man-made articulated objects.
>
>
> > … claims of the paper would be significantly stronger if the authors were showcasing that their model can also operate in more challenging setups.
>
> Although extending our research to non-rigid articulated objects like human body shapes is an exciting direction, we are specifically interested in man-made articulated objects in this study, as we have stated in the abstract and the introduction. Also, we believe learning unsupervised part decomposition for man-made articulated objects is a more challenging task than it might seem to be. And It imposes distinct and unique challenges to the community. In fact, there exist numerous discriminative supervised part segmentation and pose estimation approaches specifically targeting man-made articulated objects [1][2][3], like ours.
>
> Because man-made articulated objects impose distinct and unique challenges, our contribution is tailored for those categories. In part pose representation, we propose to structurally represent part pose using joint parameters, and we estimate motion types of parts: revolute or prismatic types. In shape representation, we propose to use a non-primitive-based representation because it can capture the non-zero genus shape of man-made articulated objects, as shown in Figure 4, which is hard to represent by primitive-based part decomposition methods. While human body shapes considered in the D-FAUST dataset are typically homeomorphic to zero-genus shapes, and each part pose can be simply represented as a $SE(3)$ matrix.  We believe these characteristics in man-made articulated object categories are already challenging, if not more difficult than human body shape. In Table 2, we will show in the updated draft that man-made articulated objects are a challenging target even for Neural Parts [5] (label IoU is 74.31%, and ours 80.99%), which claims to be applicable to the D-FAUST dataset.
>
> > … I think the current dataset is quite simple.
>
> We believe we have tested our method on the challenging dataset; we are using the same categories from the same dataset used in the recent discriminative supervised part segmentation and pose estimation study [1], and variation and number of categories we have tested is comparable with the previous works [3][4].
>
>
> > Showing additional examples, would only make the claims of the paper stronger.
>
> We will include visualization of more samples in the updated draft’s appendix.

---

> > ### Author Response · Authors · 2021-11-19
> > **Response to reviewer#2 (2/3)**
> >
> > > Can the authors describe their intuition for the second term of Eq. 2.
> >
> > We found the second term in Eq. 2 is essential to stabilize the training process. We believe the idea is similar to the standard multi-resolution training scheme seen in numerous generative studies to avoid local minima. For example, a conceptually close loss term $\mathcal{L}_{coverage}$ is used in the recent work NDG [6]. It simultaneously learns coarse human body shape by the aforementioned loss term besides the standard reconstruction loss for shape details. We attribute the effect of the second term in our approach that Eq. 2 alone makes it difficult for the gradient to reach $F^{s, c}$ and slower to learn than $F^{s, z}$. By using the second term, $F^{s, c}$ learns a reasonable category-common shape quickly, and being able to condition $F^{s, z}$ correctly allows for stable learning. In the appendix, we will add the result of the ablation study not using the second term. We have seen substantial performance degradation in part segmentation (drop by 37.2%).
> >
> >
> > > I am wondering whether the use of the VQ-VAE is really necessary.
> >
> > We have conducted the ablation study to confirm the effectiveness of using multiple constant vectors to encode canonical joint parameters and will be included in the updated draft’s appendix. Without using the multiple constant vectors, we see the performance drop by 16.2%. We have also measured the ratio of the number of constant vectors used for each category, and we see that multiple constant vectors are used. We will also include this result in the appendix.
> >
> >
> > > … the authors state that their model first tries to reconstruct the target shape with a single part and as the training progresses … Can the authors provide some intuition regarding why this is the case?
> >
> > We believe that the difference comes from the different part shape representations. Because 3D locations outside the ground truth shape are dominant, especially objects with small volumes like eyeglasses, the model first suppresses occupancy values at all 3D locations under the isosurface threshold at the beginning of the training. As training proceeds, occupancy values of 3D regions with a higher probability of being included in a GT shape (i.e., regions around the fixed type part and around rotation anchor point where displacement by different pose is smaller) exceed the isosurface threshold first. This makes it look like that the single part is first reconstructed. As the training progresses and the model learns reasonable part pose estimation, the model estimates larger occupancy values for dynamic part type in the regions with large displacement by different poses. Thus it looks like another part is reconstructed one after another as training proceeds. In the case of primitive-based approaches targeting non-articulated shapes, the region inside the primitive always exceeds the isosurface threshold and is visible from the beginning. Since the parts are located in similar positions across instances in the case of non-articulated objects, it spreads the primitives first around the target shape.
> >
> > > … what is the incentive for utilizing multiple components in later training stages?
> >
> > Although we have described the incentive in paragraph 3 of Section 3.1 in the first draft, let us explain once again for better understanding. In our method, a single part cannot capture the variety of target shapes due to their local part poses. This is because we regularize the instance-dependent shape decoder's output shape from deviating too much from the output shape of the category-common decoder by Eq. 1. Therefore, if the model does not decompose the target shape into multiple parts, it needs to reconstruct the target shape by:
> > (1) A single part shape learned by the category-common shape decoder which only takes a constant latent vector and always returns the same shape,
> > (2) The pose of the single part as a global pose,
> > (3) Residual shape learned by the instance-dependent shape decoder whose shape is constrained to the output shape of the category-common decoder.
> > With the global pose transformation and the residual shape of a single part, it is difficult to represent the shape variations due to local pose variations of multiple parts of the target shape. This incentivizes the model to use multiple parts to represent the target shape variations.

---

> > > ### Author Response · Authors · 2021-11-19
> > > **Response to reviewer#2 (3/3)**
> > >
> > > > … Since the proposed method performs worse than all baselines … I believe the authors should try to describe why this is the case.
> > >
> > > We would like first to clarify that our approach outperforms BAE-Net shape reconstruction accuracy in F-score and chamfer L1 distance, although we recognize BAE-Net’s primary focus is not accurate shape reconstruction like our work. We also demonstrate that our reconstruction is qualitatively better than BAE-Net in Figure 3. We will add the limitation section in the appendix to discuss the reconstruction accuracy gap compared to the other baselines. There are several possible reasons for the performance gap in shape reconstruction. The first reason is that our method relies on pose estimation for shape reconstruction. Although our method estimates faithful part pose as evaluated in Section 4.2 and Appendix F, even the small pose error affects shape reconstruction scores, especially for the dynamic part type.
> > > Moreover, our method regularizes instance-dependent shape decoder’s output as described in paragraph 3 of section 3.1. This regularization results in less accurate and thicker shape reconstruction of parts than the SoTA primitive-based shape representations. Nonetheless, our focus is to realize consistent part parsing based on faithful part pose estimation rather than pursuing the SoTA shape reconstruction accuracy. Improving reconstruction accuracy can be future work.
> > >
> > >
> > > > what is the number of ground-truth parts for each object category (on average)?
> > > We use 4, 3, 2, 2, 2 parts for the drawer, eyeglasses, oven, laptop, and washing machine category, respectively. We believe these numbers are comparable with the other decomposition studies (e.g., BSP-Net and BAE-Net use 4 GT parts at most).
> > > We will add the number of parts of each category in Table 7 in the appendix of the updated draft. As noted in Section 4 on the dataset, each category has a fixed number of parts, which follows the setting of [1].
> > >
> > >
> > > > … I think it would be better to fix the number of parts for all baselines …
> > >
> > > We will add an ablation study that uses the same number of primitives (=8) for all the baselines in the appendix. We found the proposed method still achieves best part segmentation performance in this setting as well. We found coarse abstraction does not necessarily lead to better part parsing. We have conducted the ablation studies for BSP-Net only using a maximum of 8 primitives, and it performs significantly worse than using a maximum of 256 primitives (Appendix E2).
> > >
> > >
> > > > … For the experiment in Table 4, can the authors clarify how is the F-score computed? ... how many points are used for computing the Chamfer-L1 loss?
> > >
> > > F-score is computed as follow:
> > > \begin{equation}
> > > {\rm Fscore} = \frac{2 * precision_1 * precision_2}{precision_1 + precision_2}
> > > \end{equation}
> > > Precision1 and 2 are mean per point Euclidean Chamfer distance accuracy thresholded by 0.01 between surface point clouds from source to target and vice versa. We use the shared code similar to CvxNet [7] for the actual calculation. For both F-score and Chamfer distance, we sample 100k points, following OccNet’s [8] protocol.
> > >
> > >
> > > > Is there any reason why the authors are not reporting the mean-IoU ...
> > >
> > > The main reason we first did not use mean-IoU is based on an argument from [9] that low to mid-range IoU (e.g., below 0.59 mentioned in [9]) is not a reliable metric. And most of the F-score values in our experiments are in the low to mid-range (Table 4 in the original draft). Please note that we found a few implicit primitive-based works that only use point-to-point distance-based metrics (F-score, L1 Chamfer distance, earthmover distance) such as BSP-Net and SIF [10]. Nonetheless, we will include mean-IoU to make the evaluation protocol compatible with previous works.
> > >
> > >
> > > > … compare the proposed method with the more recent work
> > >
> > > We will add the comparison with Neural Parts in the updated draft. They use fewer primitives than ours, yet our approach outperforms part segmentation accuracy (mean label IoU is 74.31%, and ours 80.99%).
> > >
> > >
> > > >  … clarify how the authors compute the number of learnable parameters
> > > We count the learnable parameters of the networks. We exclude the GAN discriminator of our model because it is not used in reconstruction. We use pytorch for implementation of our approach and we show the snippet calculating the learnable parameters below:
> > > ```
> > > def count_parameters(model: torch.nn.Module) -> int:
> > >     return sum(p.numel() for p in model.parameters() if p.requires_grad)
> > > ```
> > > The above snippet is based on the code from the official implementation of OccNet [8]. As an additional modification, I made sure only it counts learnable parameters. We will replace the number of the learnable parameters with the number of primitives as suggested.

---

> > > > ### Author Response · Authors · 2021-11-19
> > > > **Reference**
> > > >
> > > > [1] Xiaolong Li, He Wang, Li Yi, Leonidas J Guibas, A Lynn Abbott, and Shuran Song. Category-level articulated object pose estimation. In Proceedings of the IEEE/CVF Conference on Computer Vision and Pattern Recognition (CVPR), pp. 3706–3715, 2020.
> > > >
> > > > [2] Fanbo Xiang, Yuzhe Qin, Kaichun Mo, Yikuan Xia, Hao Zhu, Fangchen Liu, Minghua Liu, Hanxiao Jiang, Yifu Yuan, He Wang, et al. Sapien: A simulated part-based interactive environment. In Proceedings of the IEEE/CVF Conference on Computer Vision and Pattern Recognition (CVPR), pp. 11097–11107, 2020.
> > > >
> > > > [3] Ben Abbatematteo, Stefanie Tellex, and George Konidaris. Learning to generalize kinematic models to novel objects. In Proceedings of the Conference on Robot Learning (CoRL), pp. 1289–1299, 2020.
> > > >
> > > > [4] Frank Michel, Alexander Krull, Eric Brachmann, Michael Ying Yang, Stefan Gumhold, and Carsten Rother. Pose estimation of kinematic chain instances via object coordinate regression. In Proceedings of the British Machine Vision Conference (BMVC), pp. 181.1–181.11, 2015.
> > > >
> > > > [5] Despoina Paschalidou, Angelos Katharopoulos, Andreas Geiger, and Sanja Fidler. Neural Parts: Learning Expressive 3D Shape Abstractions With Invertible Neural Networks.  In Proceedings of the IEEE/CVF Conference on Computer Vision and Pattern Recognition (CVPR), pp. 3204–3215, 2021.
> > > >
> > > > [6] Aljaz Bozic, Pablo Palafox, Michael Zollho ̈fer, Justus Thies, Angela Dai, and Matthias Nießner. Neural deformation graphs for globally-consistent non-rigid reconstruction. In Proceedings of the IEEE/CVF Conference on Computer Vision and Pattern Recognition (CVPR), pp. 1450–1459, 2021.
> > > >
> > > > [7] Boyang Deng, Kyle Genova, Soroosh Yazdani, Sofien Bouaziz, Geoffrey Hinton, and Andrea Tagliasacchi. Cvxnet: Learnable convex decomposition. In Proceedings of the IEEE/CVF Conference on Computer Vision and Pattern Recognition (CVPR), pp. 31–44, 2020a.
> > > >
> > > > [8] Lars Mescheder, Michael Oechsle, Michael Niemeyer, Sebastian Nowozin, and Andreas Geiger. Occupancy networks: Learning 3d reconstruction in function space. In Proceedings of the IEEE/CVFConference on Computer Vision and Pattern Recognition (CVPR), pp. 4460–4470, 2019.
> > > >
> > > > [9] Maxim Tatarchenko,  Stephan R Richter,  Ren ́e Ranftl,  Zhuwen Li,  Vladlen Koltun,  and ThomasBrox.  What do single-view 3d reconstruction networks learn?  In Proceedings of the IEEE/CVFConference on Computer Vision and Pattern Recognition, pp. 3405–3414, 2019.

---

### Official Review · Reviewer_bBs4 · 2021-11-02

**Correctness:** 4
**Technical Novelty And Significance:** 3
**Empirical Novelty And Significance:** 3
**Recommendation:** 8
**Confidence:** 4

**Main Review:**

# Strengths:
The proposed method attacks the very challenging problem of capturing the shape of an articulated object, together with its variations within the object category, its decomposition into parts in a parsimonious way, and the estimation of each part's pose in terms of a joint state and, in the case of revolute joints, a pivot point.

Although the overall architecture is somewhat complicated with numerous loss terms involved, the proposed method achieves improved performance with respect to other unsupervised generative part decomposition methods on standard metrics considered for the task. In terms of part pose estimation, it achieves comparable performance to the NPCS which assumes part segmentation supervision, even though the proposed method is unsupervised. Additionally, the part parsing produced is more parsimonious and intuitive qualitatively, in terms of interpretability.

The experimental evaluation, although compact, is quite comprehensive covering many aspects of the method. It considers also a different input modality (depth maps) as well as the application of the method on real data. Also, the authors have made significant effort to assure that comparison with other baselines is fair.

Regarding reproducibility, all the implementation details are provided, making it possible to reproduce the results. It would also be important though to make the source code available.

# Weaknesses:
There are few weaknesses I can identify. One is that the notation is quite heavy, involving numerous symbols and many superscipts/subscripts. Admittedly, given the overall complexity of the task and the method proposed, it might be difficult to make the notation lighter. One thing that might help, would be to include full words in the subscripts of the loss terms, as their number is high and it is not easy to remember what each of them tries to achieve. Nevertheless, given the compactness of the Methods section, the paper manages to provide a quite clear explanation of the method.

A minor issue, in my view, is that the fact that a discriminator is used is stated quite late in the paper (Section 3.2, pg. 6). I would prefer to know earlier (possible from the introduction), that the method is based on an adversarial learning setting.

Another issue of the overall presentation regards the hyperparameters. Some of them, like e_i and y_i, provide crucial information to make learning possible and seem to depend to some extent on the structure of the particular objects, in general, and the way they are represented in the dataset, specifically. I would prefer if these hyperparameters were presented and discussed in a more straightforward way, rather than been less evident within the text.

## Minor Comments
- Introduction: "have mainly focus"
- Figure 1: Left and Right labels should be swapped
- Notation: \mathcal{X} is not properly defined, this makes difficult to interpret the domain of \hat{O}_i (R^3\times \mathcal{X})
- The phrase: "Instead of using a single constant vector, ... biases" (before eq.4 on pg.5) is not very clear
- Tables: please indicate which metrics are higher-better and which are lower-better
- Appendix B: Wrong citation for NPCS

**Summary Of The Paper:**

The paper presents a method for unsupervised generative part decomposition based on part kinematics. By employing a common encoder, operating on the input point cloud, the method learns a non-primitive-based implicit field for each part using a set of category and instance specific shape decoders, and the joint parameters of each part using a set of category and instance specific pose decoders.

**Summary Of The Review:**

Overall, I think that the paper is quite strong and the method proposed contributes some quite interesting ideas for effectively addressing the challenging problem of unsupervised pose-aware generative part decomposition. The experimental evaluation is comprehensive and follows correct practices.

---

> ### Author Response · Authors · 2021-11-19
> **Response to reviewer#1**
>
> We appreciate reviewer#1’s encouraging comment, deep understanding of our contributions, and constructive suggestions.
>
>
>
>
> >  I would prefer to know earlier … that the method is based on an adversarial learning setting.
>
> We will state that we use adversarial learning as one of the regularizers in the introduction in the updated draft. We put it this way rather than putting a strong emphasis on adversarial learning to avoid a wrong impression for the readers. As shown in Table 5 in the first draft, it works as one of the regularizers rather than the primary source of supervision. And the main source of supervision comes from shape reconstruction loss of an autoencoder framework.
>
>
> > … I would prefer if these hyperparameters were presented and discussed in a more straightforward way ...
>
> We will visualize ${\bf e}_i$ and also correspondence between ${\bf e}_i$ to $y_i$ in the appendix of the updated draft.
>
>
> > … notation is quite heavy ...
>
> We will improve the notation as time allows. We are currently modifying loss terms’ subscripts to a full name if tight space constraints allow.

---

### Author Response · Authors · 2021-11-19
**Our General Response**

We thank all reviewers for their comments. We will revise our current draft accordingly and upload the updated draft soon.

We apologize that we have found a bug in our code for aggregating the F-score in Table 4 and Table 3 of the first draft. The scores have slightly changed for all the items except the drawer category of the proposed method. The F-score was misprinted from mean-IoU, and we have already updated the draft with fixed numbers. The conclusion on the reconstruction evaluation is still the same as before; BSP-Net and NSD outperform ours, and ours has superior performance than BAE-Net.

---

### Author Response · Authors · 2021-11-23
**General comments on revision**

We would like to thank all the reviewers for the suggestions and questions to improve our paper.

### Major revision

1. Added the new baseline, Neural Parts [Paschalidou+ 2021] ($\textcolor{red}{R2}$, $\textcolor{green}{R3}$, $\textcolor{magenta}{R4}$).
2. Eased claim on generative applications ($\textcolor{red}{R2}$): We eased the claim for generative applications (shape editing, novel shape generation) in the draft. In the introduction, we also re-emphasize that we aim to learn consistent part parsing as a shape abstraction approach for man-made articulated objects in Section 4.3. On the interpolation result in Figure 5, we also eased the conclusion on applicability to novel shape generation. We limit the scope to verifying the desired disentanglement between shape and pose. To emphasize our contribution on part parsing rather than shape reconstruction accuracy, we modified Figure 4 by adding visualization of segmentation result on GT mesh shape.
3. Restructured method section ($\textcolor{red}{R2}$): We made the two significant changes in the method section; (1) we switched the subsections on the shape and the pose representations, and (2) we made an independent subsection for the losses.
4. Added additional visualization of part segmentation and mesh reconstruction on various samples in Figure 10 in the appendix ($\textcolor{red}{R2}$).
5. Added new ablation studies ($\textcolor{red}{R2}$): (1) In Appendix H, we added the ablation study results on the effect of VQ-VAE like multiple constant vectors and the second term of Eq. 2 of the main paper, (2) In Appendix E3 and Table 9, we added the ablation study on part parsing with an aligned number of parts and primitives.
6. Modified Section 4.3 on shape reconstruction and added analysis on shape reconstruction result in Appendix G ($\textcolor{red}{R2}$): In Section 4.3, to avoid confusion that our approach tries to claim accurate shape reconstruction like NSD, BSP-Net, we specifically compare our result with the closes baseline BAE-Net whose focus is also generative part parsing using non-primitive-based shape representation, rather than pursuing reconstruction accuracy. We also analyzed the performance gap against the other baselines (NSD, BSP-Net, and Neural Parts) by adding qualitative evaluation in Appendix G.
7. Added volumetric IoU as additional shape reconstruction metrics ($\textcolor{red}{R2}$).

### Minor revision

1. In the introduction, we added an explanation on how we tackle the challenges raised in the third paragraph in the penultimate paragraph ($\textcolor{red}{R2}$).
2. Reported the number of ground-truth parts in Table 7 in the appendix ($\textcolor{red}{R2}$).
3. Moved Figure 3 from the appendix on visualization of the joint parameters ($\textcolor{red}{R2}$).
4. Added high-level explanations of each component in the pipeline in the second paragraph of Section 3 and the brief version in Figure 2 due to the space constraints ($\textcolor{red}{R2}$).
5. Added a description of semantic capability in the last paragraph of the introduction ($\textcolor{magenta}{R4}$).
6. Added a sentence on the usage of adversarial losses in the penultimate paragraph of the introduction ($\textcolor{blue}{R1}$).
7. Explicitly defined $\mathcal{X}$ in the first paragraph of Section 3 ($\textcolor{blue}{R1}$).
8. Improved displays in tables ($\textcolor{blue}{R1}$, $\textcolor{red}{R2}$, $\textcolor{magenta}{R4}$): We show arrows indicating the metrics' direction and added descriptions in the captions ($\textcolor{blue}{R1}$). We explicitly state which metric is used in the table ($\textcolor{red}{R2}$). And we highlighted the number in Table 4 against the directly compared baseline, BAE-Net ($\textcolor{magenta}{R4}$). For Tables 3 and 6, we kept the current tables since we show the supervised baseline as a reference and do not emphasize superiority on the input modality.
9. Fixed typos, grammatical errors, etc., reported from all the reviewers.

###3 Remaining edit

Complicated symbol notations and subscript of losses ($\textcolor{blue}{R1}$, $\textcolor{green}{R3}$): Replacing symbols in a sentence requires time for careful work. Therefore, we would like to work on this in camera-ready if accepted to avoid confusing the reviewers due to unexpected typos caused by working under time constraints.

We would like to work on the following changes in camera-ready if accepted due to the time constraint in the rebuttal period.
1. Display hyperparameters in a more straightforward way ($\textcolor{blue}{R1}$)
2. Visualize category-specific shapes ($\textcolor{magenta}{R4}$)
3. Ablation on predefined part numbers ($\textcolor{magenta}{R4}$)

---

### Decision · Program_Chairs · 2022-01-20

**Decision:**

Reject

**Comment:**

The submission received split reviews: two reviewers recommended accepts, and the other two rejects. The AC went through the reviews, responses, and discussions carefully. The AC appreciates the authors' effort during the response period and agreed that the revision has addressed some of the concerns of the reviewers. However, a few key issues are not fully addressed. This includes results on additional, more complex object categories; discussion on why the performance of the proposed method is not even as comparable as BSP-Net (Table 4); and others.  Further, while the authors have significantly refactored the paper to address the concern on presentation and clarity, the changes are too major for the reviewers to review during the response period (the reviewers are expected to check minor updates, but not review a new paper during the response period).

Considering all pros and cons, the AC recommends rejection. The authors are encouraged to revise the paper for the next venue.